# The population genetics of convergent adaptation in maize and teosinte is not locally restricted

Silas Tittes[1,2,3]*, Anne Lorant[4], Sean P McGinty[5], James B Holland[6,7], Jose de Jesus Sánchez-González[8], Arun Seetharam[9], Maud Tenaillon[10], Jeffrey Ross-Ibarra[1,3,11]*

[1]Department of Evolution and Ecology, University of California, Davis, Davis, United States; [2]Institute of Ecology and Evolution, University of Oregon, Eugene, United States; [3]Center for Population Biology, University of California, Davis, Davis, United States; [4]Department of Plant Sciences, University of California, Davis, Davis, United States; [5]Department of Integrative Genetics and Genomics, University of California, Davis, Davis, United States; [6]United States Department of Agriculture– Agriculture Research Service, Raleigh, United States; [7]Department of Crop and Soil Sciences, North Carolina State University, Raleigh, United States; [8]Centro Universitario de Ciencias Biológicas y Agropecuarias, Universidad de Guadalajara, Zapopan, Mexico; [9]Department of Ecology, Evolution, and Organismal Biology; Genome Informatics Facility, Iowa State University, Ames, United States; [10]Génétique Quantitative et Evolution - Le Moulon, Université Paris-Saclay, INRAE, CNRS, AgroParisTech, Gif-sur-Yvette, France; [11]Genome Center, University of California, Davis, Davis, United States

*For correspondence:
silas.tittes@gmail.com (ST);
rossibarra@ucdavis.edu (JR-I)

## eLife Assessment

This **useful** study examines patterns of diversity and divergence in two closely related sub-species of Zea mays, patterns that have bearings on local adaptation in maize and teosinte at intermediate geographic scales. The authors suggest that convergent evolution has been facilitated by both standing variation and gene flow, with independent selective sweeps in the two species. While the data themselves are **solid**, there are limitations concerning population sampling, false positive rates in sweep detection and integration of phenotypic data, which make it difficult to draw definitive conclusions. The work should in principle be of broad interest to colleagues studying the relationship between domesticated species and their progenitors, as well as those studying instances of parallel evolution.

**Abstract** What is the genetic architecture of local adaptation and what is the geographic scale over which it operates? We investigated patterns of local and convergent adaptation in five sympatric population pairs of traditionally cultivated maize and its wild relative teosinte (*Zea mays* subsp. *parviglumis*). We found that signatures of local adaptation based on the inference of adaptive fixations and selective sweeps are frequently exclusive to individual populations, more so in teosinte compared to maize. However, for both maize and teosinte, selective sweeps are also frequently shared by several populations, and often between subspecies. We were further able to infer that selective sweeps were shared among populations most often via migration, though sharing via standing variation was also common. Our analyses suggest that teosinte has been a continued source of beneficial alleles for maize, even after domestication, and that maize populations have facilitated adaptation in teosinte by moving beneficial alleles across the landscape. Taken together,

our results suggest local adaptation in maize and teosinte has an intermediate geographic scale, one that is larger than individual populations but smaller than the species range.

## Introduction

As populations diverge and occupy new regions, they become locally adapted to the novel ecological conditions that they encounter. Decades of empirical work have carefully documented evidence for local adaptation, including the use of common garden and reciprocal transplant studies demonstrating that populations express higher fitness in their home environment (*Clausen et al., 1948*) as well as quantitative genetic approaches that show selection has acted on individual traits to make organisms better suited to their ecological conditions (*Savolainen et al., 2013*). It is clear from these studies that local adaptation is pervasive in natural populations.

One important but understudied aspect of local adaptation is its geographic scale. Empirical studies have documented adaptation at multiple scales, from microgeographic differentiation among mesic and xeric habitats along a single hillside (*Hamrick and Allard, 1972*) to regional (*Lowry et al., 2008*; *Whitehead et al., 2011*) and even global scales (*Colosimo et al., 2005*). A key factor determining the geographic scale of local adaptation is the distribution of the biotic and abiotic challenges to which organisms are adapting, as these features place limits on the locations over which an allele remains beneficial. Environmental features overlap with each other to varying degrees (*Tuanmu and Jetz, 2015*). The degree of overlap between environmental features may be important if mutations are pleiotropic, as an allele may not be beneficial when integrating its effect over multiple selective pressures (*Chevin et al., 2010*).

The geographic scale of local adaptation depends, too, on population structure. *Gossmann et al., 2010*, showed that the estimates of the proportion of new mutations fixed by natural selection across a number of plant species tended to overlap with zero, suggesting there is little evidence for adaptation at non-synonymous sites (though see *Williamson et al., 2014*; *Geist et al., 2019*, for more recent non-zero estimates of $\alpha$ in two plant taxa). One potential explanation raised by the authors for this surprising finding was that natural populations are often structured, such that very few adaptations would be expected to be common over the entire range of the species' distribution. Indeed, even when selective pressures are shared across populations, structure can hinder a species' adaptation by limiting the spread of beneficial alleles across its range (*Bourne et al., 2014*). Consistent with this, *Fournier-Level et al., 2011*, conducted a continent-scale survey across strongly structured populations of *Arabidopsis thaliana*, finding that alleles which increase fitness tended to occur over a restricted geographic scale. But it remains unclear if the scales identified in *Arabidopsis* are commonly shared across taxa.

The majority of local adaptation studies are motivated by conspicuous differences in the phenotypes or environments of two or more populations. As such, many instances of local adaptation that are occurring, as well as the underlying beneficial mutations being selected, may go overlooked. This hinders our ability to draw more general conclusions about the overall frequency and impact of local adaptation on a given population's evolutionary history.

Using population genetic approaches, we can compare the observed distribution of beneficial alleles across multiple populations to get a more holistic description of the history of selection. The patterns of selective sweeps and adaptive fixations that are exclusive to or shared among multiple populations can be used to measure a beneficial allele's geographic extent, which is influenced by the factors outlined above. If we infer multiple structured populations have fixed the same beneficial allele, this suggests that pleiotropy has not disrupted the adaptive value of the allele across environments or that the populations share a sufficiently similar set of selective pressures. Assessment of the relative frequency and geographic extent of unique and shared beneficial alleles thus allows us to quantify the scale of local adaptation. Additionally, when multiple populations do share an adaptive allele, we can infer the mode by which sharing occurred (*Lee and Coop, 2017*), providing further insights about the environmental and genetic context of each adaptation as well as the processes underlying allele sharing among populations.

Motivated to improve our understanding about the genetic basis of local adaptation and its geographic scale, we set out to use population genetic approaches to understand patterns of adaptation via selective sweeps in multiple discrete populations of domesticated maize *Zea mays* ssp. *mays*

and its wild relative teosinte *Z. mays* ssp. *parviglumis* growing across their native range in Mexico. *Z. mays* is an annual grass, native to southern Mexico. Maize was domesticated more than 9000 years ago (*Piperno et al., 2009*) from its common ancestor with the extant annual grass teosinte, but traditional open-pollinated populations maintain large population sizes and a surprising amount of diversity (*Bellon et al., 2018*). Maize is also the world's most productive crop (*Ranum et al., 2014*), and an important model system for genetics (*Nannas and Dawe, 2015*).

Previous work in both maize and teosinte has demonstrated clear population structure at both regional (*Pyhäjärvi et al., 2013*; *Vigouroux et al., 2008*) and fine (*Van Heerwaarden et al., 2010*; *Hufford et al., 2011*; *Takou et al., 2024*) scales, and population genetic and common garden studies in both subspecies have shown clear signatures of populations being adapted to ecological conditions across their native range. In maize this includes local adaptation to high elevation (*Gates et al., 2019*; *Janzen et al., 2021*), phosphorous (*Rodriguez-Zapata et al., 2021*), temperature (*Butler and Huybers, 2013*), and day length (*Swarts et al., 2017*). Similarly, studies of teosinte have documented local adaptation based on features such as the differential patterns of microbial community recruitment (*O'Brien et al., 2019*), elevation (*Fustier et al., 2019*; *Fustier et al., 2017*), and temperature and phosphorous (*Aguirre-Liguori et al., 2019*).

Studying local adaptation of maize and teosinte across the same geographic locations presents opportunities to disentangle multiple processes that interact with adaptation. For example, the effect of the domestication process in maize populations and their ongoing interaction and dependence on humans has created changes in the timing and types of selection imposed across all populations, as well as changes in demography (*Wright et al., 2005*). Based on population structure and differences in the abiotic environment among populations, we anticipated that local adaptation would have a small geographic scale. We predicted that sweeps would be exclusive to individual populations, and that adaptations shared between subspecies would be limited to sympatric pairs of populations growing in similar environments and with ample opportunity for genetic exchange. Because of domestication and the ongoing migration facilitated by humans, we expected that maize would show more shared adaptations, leading to a relatively larger geographic scale of local adaptation. Contrary to our predictions, our results suggest adaptations are often shared across two or more populations, and commonly between maize and teosinte. We also found that migration and standing variation have played an important role as sources of beneficial alleles, including many that are shared across the two subspecies.

## Results

We sampled teosinte (*Z. mays* subsp. *parviglumis*) individuals from six locations across its native range, along with a nearby (sympatric) population of traditionally cultivated open-pollinated maize (commonly referred to as landraces) at five of these locations (*Figure 1C*). We sampled 10 individuals from each population for each subspecies, with the exception of the Palmar Chico populations, where we took advantage of 55 and 50 individuals previously sampled for maize and teosinte, respectively (*Appendix 1—table 1*; *Chen et al., 2020*). Both Palmar Chico populations were down-sampled to 10 randomly selected individuals to facilitate comparisons to the other populations. We did, however, use a second random sample of the Palmar Chico populations to estimate the accuracy of our inference of selective sweeps (see below in Results and Materials and methods).

Rangewide samples for each subspecies were constructed by randomly selecting one individual from each population. All 195 individuals were sequenced at 20–25× coverage and aligned to version 5 of the *Z. mays* B73 reference genome (*Hufford et al., 2021*; *Portwood et al., 2019*). Analyses were based on genotype likelihoods (*Korneliussen et al., 2014*) except in cases where called genotypes were required (see Materials and methods).

### Subspecies and populations are genetically distinct despite evidence of gene flow

To assess the relationships among our sampled populations, we constructed a population-level phylogeny using Treemix (*Pickrell and Pritchard, 2012* v.1.13). As anticipated from previous work (*Buckler and Holtsford, 1996*; *Hufford et al., 2012*), we found clear divergence between two clades

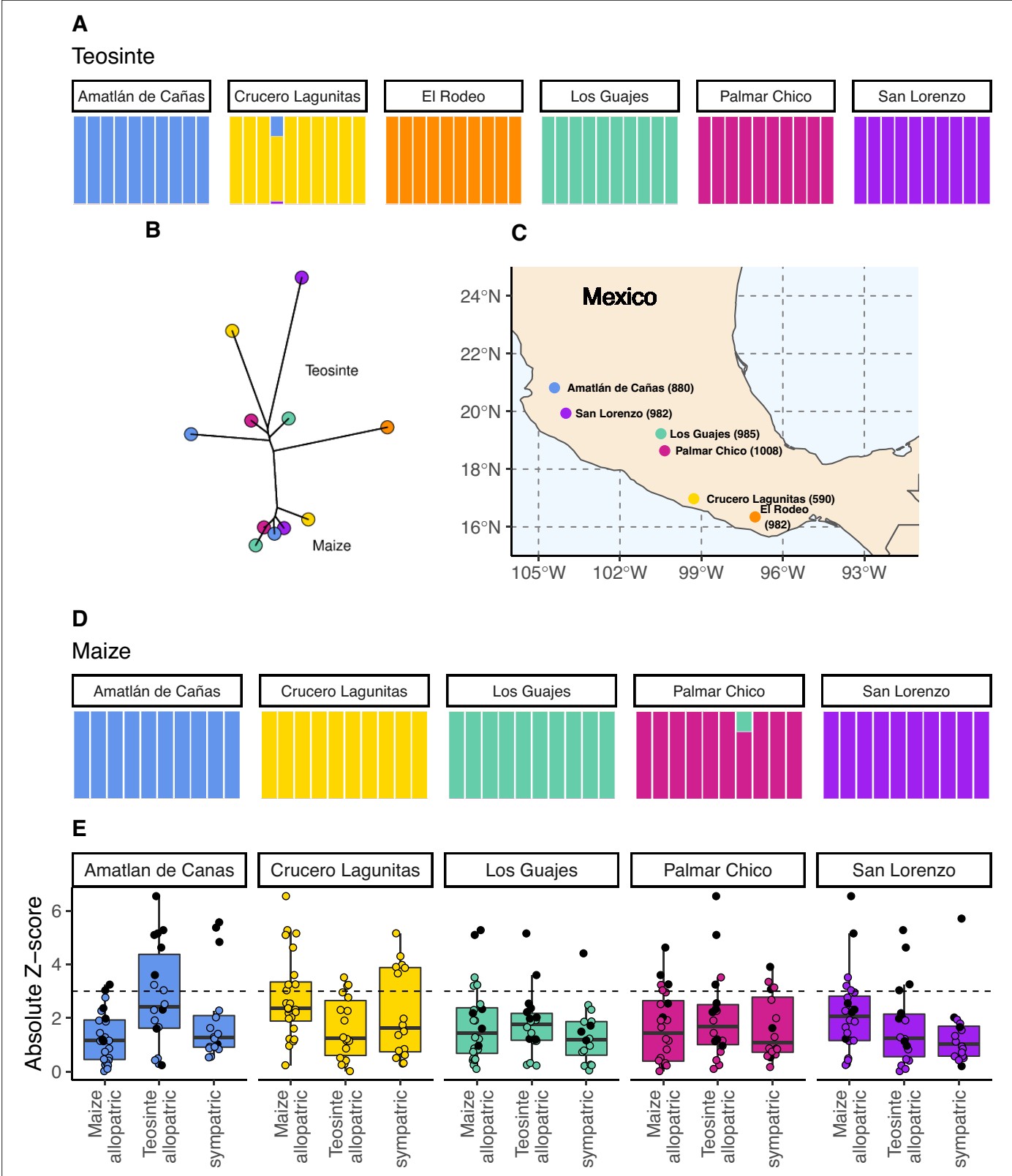

**Figure 1.** The geographic distribution, population structure, and gene flow of maize and teosinte populations. (**A** and **D**) Admixture proportions among populations within subspecies. The dominant cluster in each population is colored by sampling location. (**B**) The unrooted tree of maize and teosinte populations. (**C**) Geographic sampling locations for the studied maize and teosinte populations. (**E**) $F_4$ tests to quantify evidence of gene flow between the subspecies for allopatric and sympatric population pairs. Each point in (**E**) reports the absolute Z-score for an $F_4$ test, where a given focal population

*Figure 1 continued on next page*

*Figure 1 continued*

was partnered with another population of the same subspecies as a sister node, and two other populations from the other subspecies as a sister clade (see Materials and methods for further details). Black points show $F_4$ tests that included maize from Crucero Lagunitas, otherwise points are colored by focal population. The dotted line corresponds to our chosen significance threshold (p = 0.001).

composed of maize and teosinte populations, though the relationship among geographic locations differed between the subspecies (*Figure 1B*).

Within subspecies, populations were genetically distinct from one another. Using NGSadmix (*Skotte et al., 2013*), there was little evidence of admixture between populations of the same subspecies; only two of the sampled individuals revealed evidence of mixed ancestry (*Figure 1A and D*).

Despite the clear phylogenetic separation between the two subspecies, there is evidence for gene flow between maize and teosinte populations. We conducted $F_4$ tests using Treemix (*Pickrell and Pritchard, 2012*) with the expectation that tests including sympatric pairs would show the strongest signatures of gene flow. Instead, we found that all populations showed some evidence of gene flow with various populations of the other subspecies, as measured by the high absolute $Z$-scores of the $F_4$ statistic. We found little evidence of increased gene flow between sympatric pairs (*Figure 1E*), but $Z$-scores were sensitive to the specific combinations of non-focal populations included in each test (see Appendix 2). Specifically, we found that elevated $F_4$ tests almost always included the maize population from Crucero Lagunitas ($p < 2 \times 10^{-10}$), which was true whether or not the $F_4$ test included its sympatric teosinte. Finding higher evidence of gene flow when maize from Crucero Lagunitas is included in the test is interesting in combination with results from selective sweeps, where it also plays an outsized role (see below).

## Populations vary in their diversity, demography, and history of inbreeding

We estimated pairwise nucleotide diversity ($\pi$) and Tajima's $D$ in non-overlapping 100 kb windows along the genome in our sampled populations using ANGSD (*Korneliussen et al., 2014*). For all populations, $\pi$ was in the range of 0.006–0.01, consistent with both previous Sanger (*Wright et al., 2005*) and short-read (*Hufford et al., 2012*) estimates for both subspecies. Variation in Tajima's $D$ and $\pi$ was greater among populations of teosinte than maize (*Figure 2A and B*).

We independently estimated the demographic history for each population from their respective site frequency spectra using mushi (*DeWitt et al., 2021* v0.2.0). All histories estimated a bottleneck that started approximately 10,000 generations ago (assuming a mutation rate of 3×10⁻⁸; *Clark et al., 2005*; *Figure 2E*).

Teosinte is a primarily outcrossing grass (*Hufford, 2010*), and regional maize farming practices promote outcrossing as well (*Bellon et al., 2018*). To validate our estimated demography and characterize the history of inbreeding in each population, we compared the empirical quantiles of homozygosity by descent (HBD) segments inferred using IBDseq (*Browning and Browning, 2013*) to those simulated under the demography of each population. With the exception of the smallest HBD segments, which are more prone to inaccurate estimation, the simulated quantiles generally resemble the empirical quantiles (*Figure 2D*). This indicates that the inbreeding history of our population is adequately captured by the demography. However, consistent with previous studies of teosinte (*Hufford, 2010*), we do see variation in the distributions of HDB among populations. For example, the size distribution of HBD segments in San Lorenzo and Los Guajes were consistently larger than those simulated from their demographies, particularly for the smallest segments. This likely reflects inbreeding caused by demographic changes, particularly those further in the past that may not be as accurately captured by our demography inferences. These results are consistent with previous studies that found evidence for historical inbreeding in teosinte, particularly in individuals sampled from San Lorenzo (*Pyhäjärvi et al., 2013*). Lastly, we estimated inbreeding coefficients ($F$) using ngsRelate (*Hanghøj et al., 2019*). Although inbreeding coefficients were as high as 0.37, the mean value of $F$ was 0.017±0.001 (SE) and 0.033±0.001 (SE) for maize and teosinte (respectively) (*Figure 2E*). These values are consistent with prior estimates of the rate of outcrossing of ≈3% in teosinte (*Hufford et al., 2011*) and suggest there has been relatively little inbreeding in either subspecies in the recent past.

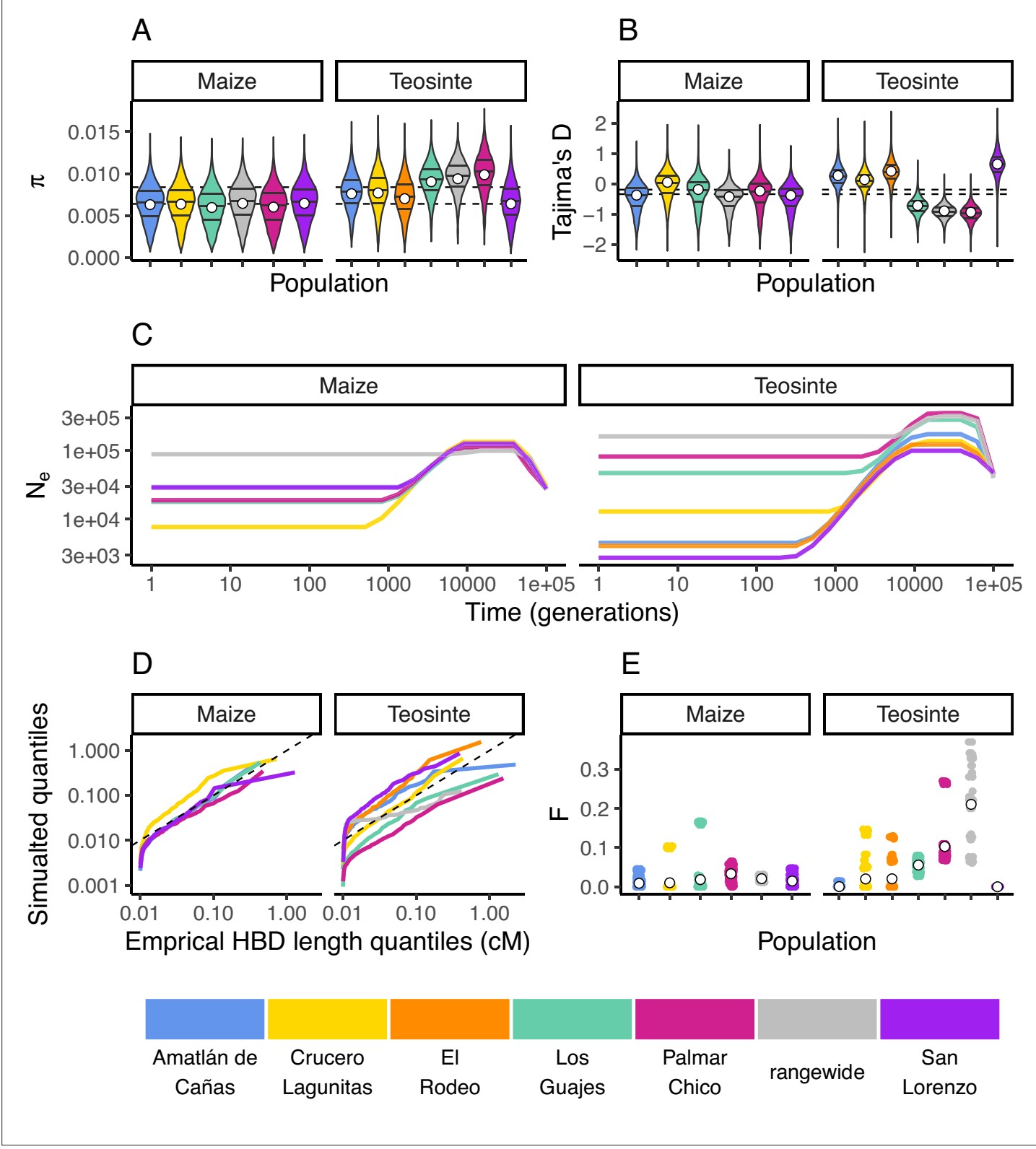

**Figure 2.** Inbreeding, diversity, and demography. The distribution of π (**A**) and Tajima's *D* (**B**) calculated in 100 kb windows for maize and teosinte populations. Dashed lines show the median values for the two subspecies. Filled white points show the median values generated from coalescent simulations under the demographic history inferred for each population. Colors for each population are as in *Figure 1* and are shown at the bottom of the figure. (**C**) The inferred demography for each population. (**D**) The quantile of observed homozygosity by descent (HBD) lengths (cM) versus

*Figure 2 continued on next page*

*Figure 2 continued*

those simulated under each population demography. Dashed lines shows the 1:1 correspondence between the axes. (**E**) The distribution of inbreeding coefficients in each population. Filled white points are the average values for each population.

## Rangewide estimates of the proportion of mutations fixed by natural selection ($\alpha$) are commensurate with that of individual populations

If populations are relatively isolated and adaptation occurs primarily via local selective sweeps, then we expect that most adaptive fixations will happen locally in individual populations rather than across the entire species range. This pattern should become even stronger if alleles experience negative pleiotropy, such that they are only adaptive in one environment and deleterious in others, further inhibiting the ability of such alleles to fix rangewide. If adaptation via sweeps is commonly restricted to individual populations, using a broad geographic sample to represent a population could underestimate the number of adaptive substitutions that occur (*Gossmann et al., 2010*). To test this, we estimated the proportion of mutations fixed by adaptive evolution ($\alpha$) (*Smith and Eyre-Walker, 2002*) across all of our populations and the rangewide samples for both subspecies. We estimated $\alpha$ jointly among all populations by fitting a nonlinear mixed-effect model based on the asymptotic McDonald-Kreitman (MK) test (*Messer and Petrov, 2013*). Across populations, $\alpha$ varied between 0.097 (teosinte San Lorenzo) and 0.282 (teosinte Palmar Chico), with more variation among teosinte populations (*Figure 3*). In contrast to our expectations, rangewide estimates of $\alpha$ were commensurate with individual populations. We additionally evaluated estimates of $\alpha$ for specific mutation types, which has been shown to be lower at sites mutating from $A/T$ to $G/C$, due to the effects of GC-biased gene conversion in *Arabidopsis* (*Hämälä and Tiffin, 2020*). While we do find some evidence that $\alpha$ predictions varied by mutation type (see *Appendix 3—figure 1*), the patterns are the opposite of that found in *Arabidopsis*, perhaps because of the increased level of methylation in maize and the higher mutation rate at methylated cytosines. Even after accounting for differences among mutation types, rangewide values remained commensurate with that of the populations.

## Teosinte populations have a higher proportion of private sweeps

Our inferences of $\alpha$ are based on substitutions at non-synonymous sites. The functional space for selection to act on occurs over many other parts of the genome besides protein coding bases, especially for large repetitive plant genomes (*Mei et al., 2018*). To identify signatures of adaptation occurring anywhere in the genome, we used RAiSD (*Alachiotis and Pavlidis, 2018*) to identify putative selective sweeps in each population, where sweep regions were identified by merging $\mu$ summary statistic outliers using a threshold defined by coalescent simulations under each population's estimated demography (see Materials and methods). Simulations suggest this approach has high precision and power compared to alternative methods over a broad range of scenarios (*Alachiotis and Pavlidis, 2018*). To further assess the accuracy of our sweep inferences, we compared the overlap in sweep regions from a second random sample from both maize and teosinte from Palmar Chico. After optimizing over a grid of hyperparameters for RAiSD (see Materials and methods), we estimated the proportion of sweeps shared between replicates to be 0.67 and 0.80 for maize and teosinte, respectively. This proportion is consistent with the false positive rates for strongly selected hard sweeps estimated from simulations we conducted under the demographic histories (see Appendix 4). As such, a non-trivial number of the putative sweep regions we inferred with RAiSD in other populations are likely also false positives, though far fewer than previously reported using alternative methods for identifying sweep regions (*Tittes et al., 2021*).

We used the inferred sweep regions to assess the degree to which adaptation is shared or locally restricted using the sweep regions we identified. We determined how many sweep regions were exclusive to one population (private), along with the number of overlapping sweep regions shared across two or more populations within and between the two subspecies. Overall, sharing was common, though fewer sweeps were exclusively shared between teosinte populations (*Figure 4A*). Across teosinte populations, 22% of sweeps were private, which was significantly greater than the 14% found in maize (binomial glm, $p = 0.0026$; *Figure 4B*). These proportions could be an overestimate because of relatively high false negative rates, causing us to count shared sweeps as unique (see *Appendix 4—figures 2 and 3* and *Appendix 1—table 1*). This likely does not impact the difference in proportions

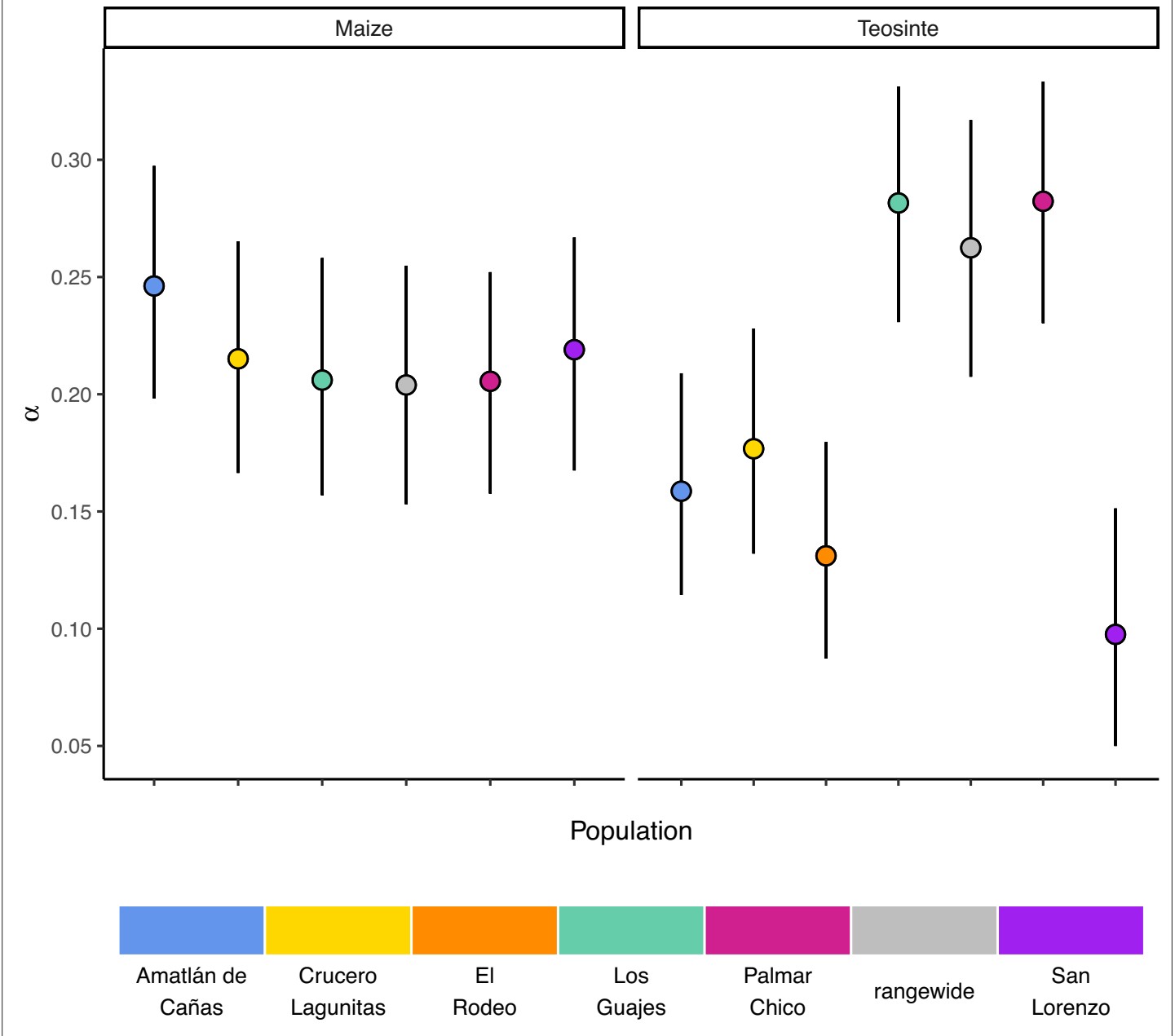

**Figure 3.** The proportion of mutations fixed by natural selection. Estimated values of the proportion of mutations fixed by natural selection (α) by population. Vertical lines show the 95% credible interval.

between species, however, since the average power to detect sweeps is similar in both subspecies (0.67 and 0.8 in maize and teosinte, respectively; see Materials and methods).

## Sympatric population pairs do not share more sweeps

If local adaptation favors certain alleles in a given environment, we might expect to see increased sharing of sweeps between sympatric populations of maize and teosinte. To look for evidence of such sharing, we used a hypergeometric test based on the number of sweeps in each population and the number of shared sweeps between population pairs, which allowed us to test if sympatric population pairs tended to have more sharing than expected by chance. In conducting this test, we incorporated our estimate of sweep accuracy (see Materials and methods). Sympatric pairs did not tend to have a lower p-value than allopatric pairs, and no population pair showed more sharing than expected by

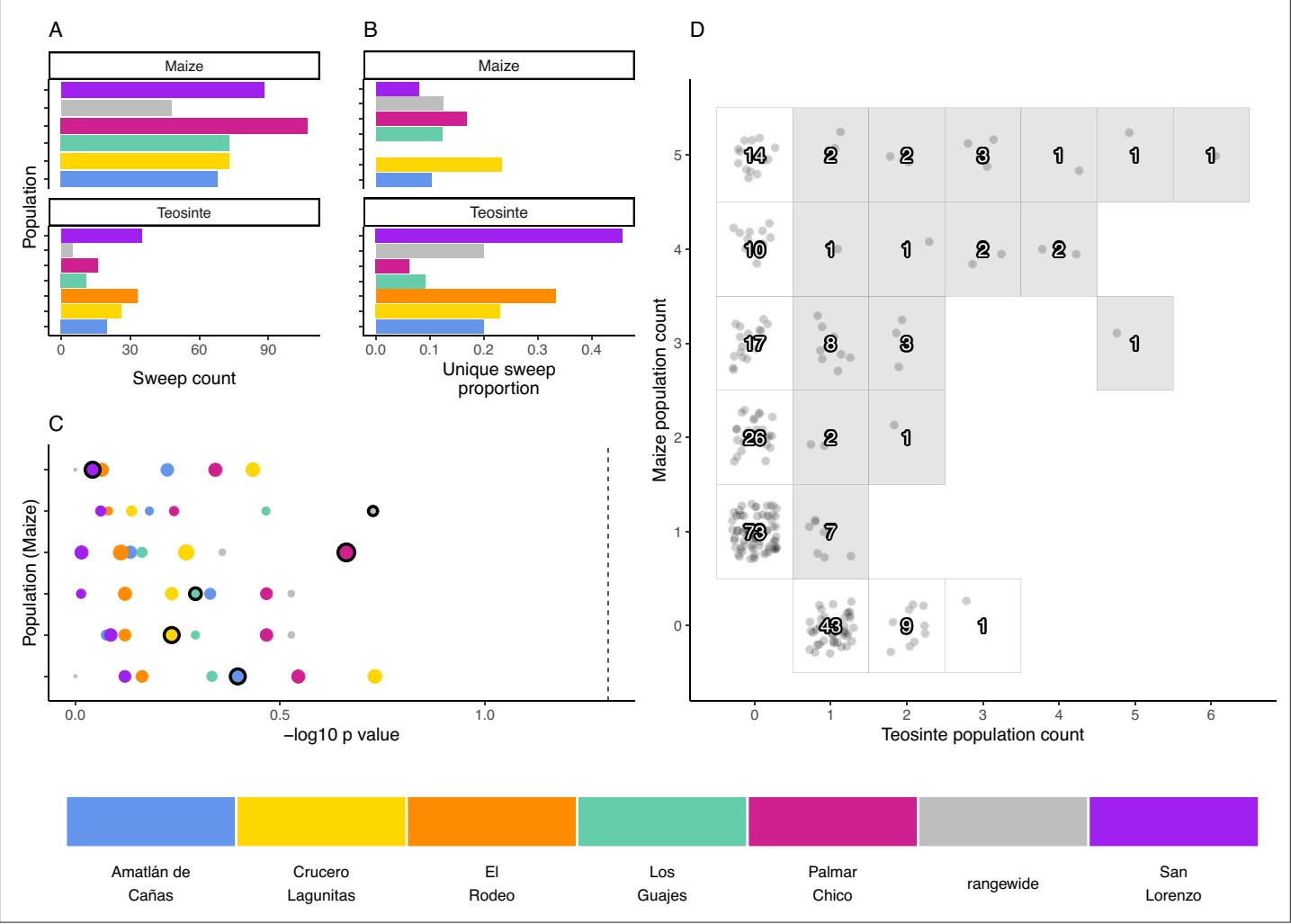

**Figure 4.** The distribution of shared and private selective sweeps. (**A**) The total number of sweeps inferred in each population. (**B**) The proportion of sweeps that are unique to each population. (**C**) Negative $\log_{10}$ p-values for hypergeometric tests to identify maize-teosinte population pairs that shared more sweeps than expected by chance (see Materials and methods). p-Values were adjusted for multiple tests using the Benjamini and Yekutieli method. Populations along the y axis are maize (order matches the legend below, with Amatlán de Cañas at the bottom), while the point color designates the teosinte population each maize population was paired with. Points with black outline highlight the sympatric population comparisons. Point size is scaled by the number of shared sweeps identified in each pair. The dotted line indicates our chosen significance level ($p = 0.05$). (**D**) Counts of shared and unique sweeps broken down by how many maize and teosinte populations they occurred in. Gray boxes show sweeps shared across the two subspecies.

chance (*Figure 4*). We additionally found that, despite evidence that sweeps are commonly shared between maize and teosinte (*Figure 4D*), there were zero sweeps exclusive to sympatric pairs; sweeps that were shared between sympatric pairs always included at least one other allopatric population.

## Convergent adaptation from migration is common among maize and teosinte populations

In instances when two or more populations shared a sweep region, we used rdmc (*Lee and Coop, 2017*; *Tittes, 2020*) to infer the most likely mode of convergence. We classified sweeps based on which composite log-likelihood model was greatest out of four possible models of convergence (independent mutations, migration, neutral, and standing variation). Of the 101 sweeps that were shared by two or more populations, there were 1, 50, and 40 sweeps inferred to be convergent via independent mutations, migration, or standing variation; and an additional 10 sweeps inferred to be neutral (*Figure 5C*). The strength of support (measured as the composite likelihood score of the best model relative to the next best) varied among sweeps and modes of convergence, but in general a single

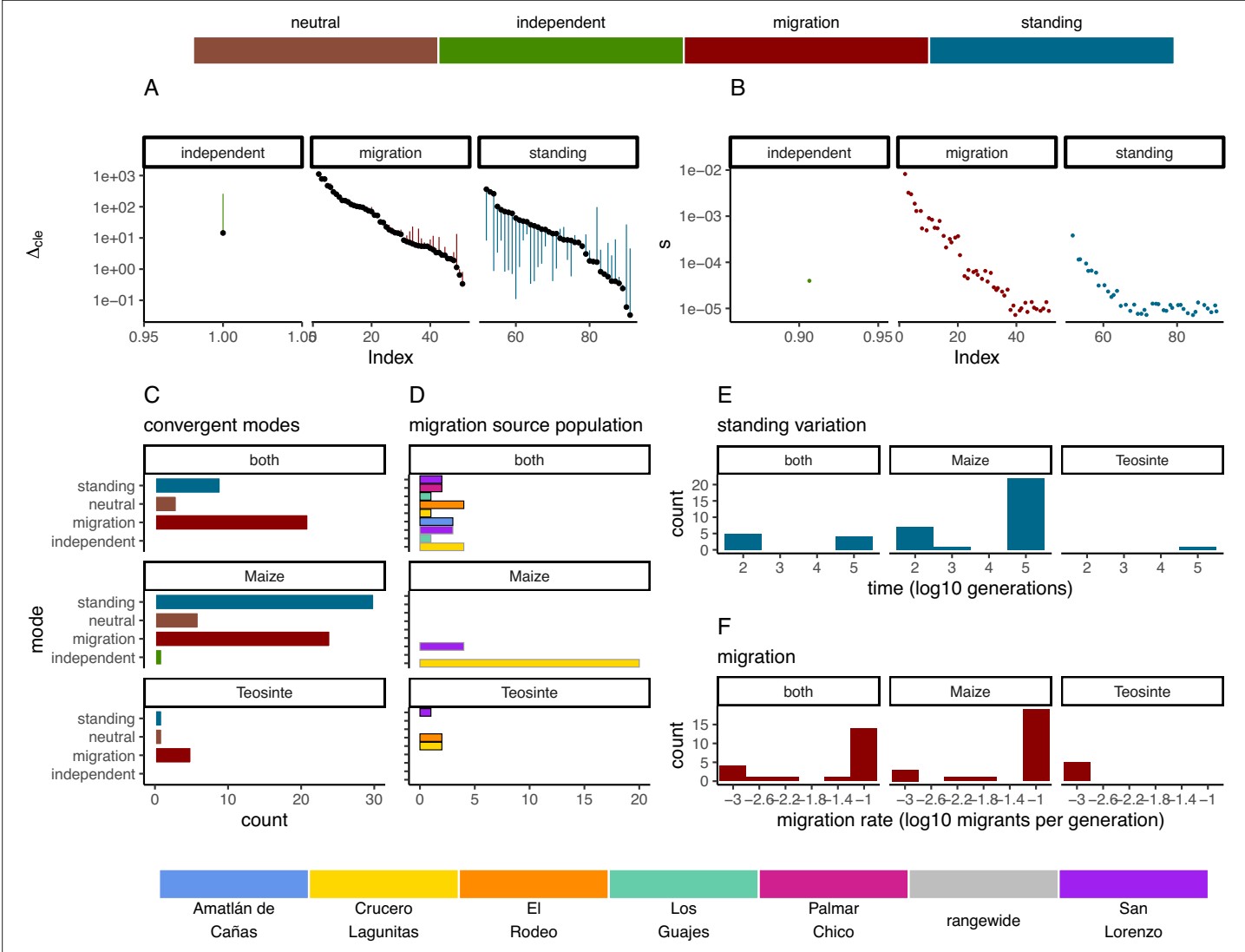

**Figure 5.** Modes of convergent adaptation and affiliated parameters for shared selective sweeps. (**A**) The difference in composite likelihood scores for the best supported mode of convergent adaptation (colors in top legend) compared to next best mode (black points), and best mode compared to the neutral model (other end of each line segment above or below black point). (**B**) Selection coefficients colored by the most likely mode of convergent adaptation. (**C**) Number of shared sweeps for both subspecies that were inferred to be from each convergent adaptation mode. (**D**) The most likely source population for shared sweeps that converged via migration. Bars are colored by population (bottom legend) and are outlined in black for teosinte and gray for maize. (**E**) Observed frequency of the inferred time in generations that each selected allele persisted prior to selection for models of convergent adaptation via standing variation. (**F**) Observed frequency of each inferred migration rate value for models of convergent adaptation via migration. (**C**, **D**, **E**, and **F**) are partitioned by which subspecies shared the sweep.

model tended to be clearly favored among the alternatives (*Figure 5A*). Further, across sites proposed to be the locus of the beneficial mutation there was a strong tendency for the next best composite likelihood score to be at a site nearest to the maximum composite likelihood site, suggesting relatively high confidence in the location of the sweep (*Appendix 4—figure 6*). Selection coefficients for sweeps varied among modes, with convergence via migration having the highest average estimate (*Figure 5B*). When migration was the mode of convergence and sweeps were shared by both subspecies, no one population stood out as the most frequent, with fairly even counts across populations of both subspecies. In contrast, convergence via migration exclusive to teosinte was seen rarely and from only three source populations, while the most common source population for convergence via migration exclusive to maize was almost entirely from Crucero Lagunitas (*Figure 5D*). We provide

hypotheses for why maize from Crucero Lagunitas would be a common source population for beneficial alleles in maize only, and how this relates to our $F_4$ tests findings (*Figure 1*) in the Discussion.

In convergence models with migration, we tested migration rates between $10^{-3}$ and $10^{-1}$. The most likely migration rate varied across sweeps, but tended to be $10^{-1}$ for the majority of sweeps shared by the two subspecies and sweeps exclusive to maize. In contrast, the lowest migration rate ($10^{-3}$) was always the most likely for sweeps exclusive to teosinte (*Figure 5F*), although there were only five instances total, suggesting convergence via migration exclusive to teosinte at any rate is rare. Together, these findings indicate that many alleles are adaptive in the genomic background of both maize and teosinte, and that adaptive alleles are commonly shared between the two subspecies, but that distinct patterns emerge in the presence or absence of maize populations sharing the signature of adaptation.

## Discussion
### Local adaptation occurs at intermediate scales
*Gossmann et al., 2010*, hypothesized that population structure within a species could limit the fixation of adaptive alleles across a species range, causing a reduction in the proportion of mutations fixed by positive selection ($\alpha$). Based on this hypothesis and the strong population structure we observed (*Figure 1*), we expected that rangewide samples would have smaller estimates of $\alpha$. Instead, $\alpha$ for the rangewide samples of both maize and teosinte were commensurate with that of individual populations (*Figure 3*), a pattern that persisted even when we considered $\alpha$ estimated from several different mutation types (*Figure 1*). However, we acknowledge that our evidence relies primarily on the patterns found in teosinte; due to a shared domestication history maize populations have less independent evolution and a larger proportion of fixed differences are shared among all populations, leading to a similar $\alpha$ estimate between individual populations and the rangewide sample. Overall our findings are inconsistent with the patterns we would expect fine-scale local adaptation to generate, where adaptive substitutions for a given population should not be shared by other populations experiencing their own distinct local selective pressures and antagonistic pleiotropy suppresses rangewide fixation in alternative environments. Our findings make sense in the light of other work studying pleiotropy's impact on adaptive evolution in *Arabidopsis*, which found that most mutations impact few traits and that the genetic architecture was largely non-overlapping when studied across multiple environments (*Frachon et al., 2017*). Our results are consistent with models in which locally beneficial alleles are simply neutral elsewhere, a process thought to be more common when overall levels of gene flow are low (*Wadgymar et al., 2022*). One possible explanation for the considerable overlap in $\alpha$ between rangewide and individual populations is that most of the fixations occur on long branches shared by all populations. This does not appear to be the case (see *Appendix 1—table 1*), likely because the coalescent within *Z. mays* ssp. *parviglumis* is a substantial fraction of the relatively recent divergence to the outgroups *Zea diploperennis* and *Zea luxurians* (*Chen et al., 2022*). Lastly, it is worth nothing that while the asymptotic MK method we employed has been shown to provide reliable estimates of $\alpha$ when fixations are due to strong beneficial mutations (*Messer and Petrov, 2013*), it assumes there is no contribution of positively selected polymorphisms, and as such, our estimates may be conservative compared to methods that account for them (*Uricchio et al., 2019*).

We found a similar pattern from our analysis of shared versus unique selective sweeps, which were more often shared by at least one other allopatric population. Similar to our predictions for $\alpha$, we expected that local adaptation would lead most sweeps to be exclusive to individual populations. Instead, the average proportion of sweeps exclusive to a single population was low to moderate for maize and teosinte populations, respectively (*Figure 4*). We also expected that maize and teosinte populations growing in close proximity would share similar local selective pressures and would therefore share more signatures of adaptation. However, no pairs showed evidence of sharing more sweeps than would be expected by chance, and overall sympatric pairs did not show increased sharing of selective sweep regions compared to allopatric pairs (*Figure 4*). This regional scale of local adaptation is consistent with patterns seen in maize adaptation to the highlands (*Calfee et al., 2021*), where sympatric maize and teosinte populations show little evidence of adaptive gene flow, yet some adaptive teosinte introgression appears widespread among highland maize.

There are a number of considerations to make in the interpretation of our results. The two methods we used to identify signatures of adaptation, estimating $\alpha$ and identifying signatures of selective sweeps, are best suited for adaptation that leads to fixation of beneficial alleles, and/or mutations of large effect. For the moderate population sizes and selection coefficients observed here, fixation of new beneficial mutations takes a considerable amount of time, on the order of $4log(2N)/s$ generations (*Charlesworth, 2020*). Compared to the sojourn time of adaptive mutations, our populations may have occupied their current locations for relatively few generations. As a result, the selective sweeps underlying local adaptation to the selective pressures that populations currently face are more likely to be incomplete, so may be more difficult to detect (*Xue et al., 2021*; *Pritchard et al., 2010*). Likewise, the adaptive sweeps that have completed may have been under selection in ancestral populations that occupied different environments than the sampled individuals, and their signatures may no longer be detectable (*Przeworski, 2002*), placing a limit on the temporal resolution with which we can make inference about instances of local adaptation. Conversely, it is possible that some of our shared sweeps represent selection that happened prior to divergence between closely related populations. This could explain why some populations (teosinte from Los Guajes and Palmar Chico) have fewer unique sweeps (*Figure 4B*), and would similarly create a mismatch in the conditions that populations currently occupy and those of the ancestral population where selection was experienced. Another complication in detecting local adaptation relates to the size and complexity of plant genomes. Large genomes may lead to more soft sweeps, where no single mutation driving adaptive evolution would fix (*Mei et al., 2018*). Like incomplete sweeps, soft sweeps are harder to identify (*Schrider and Kern, 2016*; *Pritchard et al., 2010*), which could obscure the signatures of local adaptation. Even if our populations have occupied their current locations for a sufficient duration for local adaptation to occur, the completion of selective sweeps may be hindered by changes and fluctuations in the local biotic and abiotic conditions. Relatively rapid change in local conditions could also result in fluctuating selection, such that most alleles do not remain beneficial for long enough to become fixed (*Rudman et al., 2022*).

While our focus has been on the trajectory of individual beneficial alleles, the genetic basis of many adaptive traits may be highly polygenic. Allele frequency changes underlying polygenic adaptation are more subtle than those assumed under selective sweeps, making them harder to detect (*Pritchard et al., 2010*). Evaluating local adaptation in maize and other systems will be facilitated by studying the contribution of polygenic adaptation to the evolution of complex traits. However, if adaptation across our studied populations were strictly polygenic, and especially if it were acting on alleles with small effect sizes, we would expect to find few to no shared sweeps. The fact that we find many instances of sharing across populations suggests that a non-trival amount of local adaptation is occurring via selective sweeps, or through polygenic adaptation acting on a few loci with large effects that leave a signature similar to that expected under a selective sweep model.

## Differences in diversity and demography influence adaptation in maize and teosinte

While our results were generally similar between the two subspecies and among the sampled populations, there are several important differences. The most obvious difference between the subspecies is the ongoing interaction and dependence of maize on humans via domestication and farming. Compared to teosinte, maize had lower average genome-wide estimates of diversity (*Figure 2A*). These differences are consistent with the previously discovered pattern that diversity tends to be lower in crops compared to their wild relatives (*Doebley, 1990*; *Hufford et al., 2012*), a pattern putatively driven by domestication bottlenecks (*Eyre-Walker et al., 1998*). In line with this argument, the few teosinte populations with lower diversity than those in maize (El Rodeo and San Lorenzo) were inferred to have the most substantial bottlenecks and historical inbreeding (*Figure 2*). More generally, we found that $\pi$ and Tajima's $D$ were more variable among teosinte populations, indicative of differences in their demographic histories.

Our demographic inferences suggest that all populations had signatures of a bottleneck, the timing of which coincides with the beginning of maize domestication sometime prior to ≈9000 years ago (*Piperno et al., 2009*). The severity of the bottleneck varied considerably across populations, particularly for teosinte. While finding a bottleneck in the maize populations is consistent with domestication, it is less clear why we found a similar bottleneck for the teosinte populations at

approximately the same time. One possibility is that the teosinte bottlenecks reflect land use change induced by human colonization. For example, Mesoamerican phytolith records in lake sediments show evidence of anthropogenic burning as early as 11K years BP (*Piperno, 1991*). The establishment and spread of human populations over the subsequent millennia would require an ever-increasing area for farming, dwellings, transportation, and trade (*Haines et al., 2000*). Such land use changes would likely encroach on the habitat available for teosinte and drive species-wide census size declines. Given the success of maize breeding and domestication, we anticipated a recent expansion for maize populations as previously seen (*Beissinger et al., 2016*; *Wang et al., 2017*). However, with only 10 individuals per population, recent expansions will be difficult to detect with approaches based on the site frequency spectrum (*Keinan and Clark, 2012*; *Coventry et al., 2010*). The demography of the rangewide samples for both subspecies showed little evidence of the bottleneck inferred in individual populations, likely due to the reduced sampled size (five and six individuals) for the rangewide data. We additionally used strong regularization penalties to avoid overfitting (see Materials and methods), which limits the detection of rapid and dramatic changes in population size. The near-constant size of the rangewide samples and the lack of recent expansion in maize are both likely influenced by this modeling choice.

Demographic inferences are known to be sensitive to the effects of linked selection (*Ewing and Jensen, 2016*). However, all populations independently converged to similar values in the oldest generation times, around the time when we would expect the ancestral lineages would have coalesced (*Figure 2C*). This suggests any biases in the estimated population sizes that are specific to maize, which has had recent explosive population growth, are occurring in the more recent past (*Beissinger et al., 2016*).

Differences in adaptation between maize and teosinte, and among populations, were apparent based on differences in the patterns of selective sweeps. Maize had a higher proportion of selective sweeps shared with at least one other population (*Figure 4*). The greater number of shared sweeps in maize populations is likely the result of their recent shared selective history during the process of domestication, resulting in a set of phenotypes common to all maize (*Stitzer and Ross-Ibarra, 2018*). In comparison, the higher proportion of unique sweeps in teosinte suggests local adaptation has played more of a role in shaping their recent evolutionary history. Teosinte grows untended, and did not undergo domestication, leaving more opportunity for divergence and local selection pressures to accumulate differences among populations. This is reflected in the inferred population history, which had longer terminal branch lengths for teosinte (*Figure 1B*), suggesting there is increased genetic isolation among teosinte populations due to some combination of longer divergence times, reduced gene flow, and/or higher genetic drift from reduced population sizes.

## Convergent adaptation is ubiquitous

We found convergent adaptation to be common among populations and subspecies (*Figures 4 and 5*). The frequency of convergence further suggests there are a large number of mutations that are beneficial in more than one population, even when placed in the different genomic backgrounds of the two subspecies. Our approach allowed us to distinguish between multiple potential modes of convergence, including a neutral model that models allele frequency covariance by drift alone (*Lee and Coop, 2017*). The distribution of most likely selection coefficients of the inferred beneficial alleles suggests the strength of selection is moderate to strong, though this estimate is likely biased as strong positive selection will be easier to detect. Convergence via independent mutations was by far the least frequent mode. This is consistent with previous analyses of domestication (*Hufford et al., 2012*) and adaptation (*Wang et al., 2021*) in maize, and unsurprising given evidence for ongoing gene flow (*Figure 1E*), the relatively short evolutionary time scales, and the low probability that even strongly selected new mutations can overcome drift multiple times independently.

For convergent sweeps that occurred via standing variation within maize or shared between maize and teosinte, the distribution of generation times that the selected variant was standing before the onset of selection tended to be bimodal, with both long and short standing times (*Figure 5E*). In contrast, the sweep exclusive to teosinte was inferred to be standing variation for more generations, although there was only one such instance.

Sweeps that occurred via standing variation and shared between subspecies were often found in only a subset of maize populations. Many of these sweeps likely reflect the presence of structure in

ancestral populations, suggesting different alleles beneficial to maize were likely derived from more than one teosinte population. The bimodal features of sharing seem at face value surprising. How can an allele be standing variation for so many generations after divergence but prior to selection when the populations diverged less than 10,000 generations ago? We speculate that ancestral population structure and limited sampling of six populations could explain the pattern. For example, extremely long-standing times that predate domestication may reflect divergence between our sampled teosinte populations and the populations most closely related to those that gave rise to domesticated maize. More comprehensive sampling could help to resolve patterns of local adaption.

The most common mode of convergent adaptation was via migration, and frequently occurred between geographically disparate populations (*Appendix 4—figure 4*). This included a relatively large number of shared sweeps via migration between maize and teosinte (*Figure 5* and *Appendix 4—figure 5*). There is ample evidence that maize and teosinte are capable of hybridizing (*Wilkes, 1967*; *Ellstrand et al., 2007*; *Ross-Ibarra et al., 2009*), and previous work has identified gene flow between geographically disparate populations of maize and *Z. mays mexicana* (*Calfee et al., 2021*). Further, convergence via migration between geographically disparate maize populations has been inferred during adaptation to high elevations (*Wang et al., 2021*).

Our findings on convergence via migration point to an intriguing hypothesis, namely, that some number of alleles that are beneficial to teosinte may have originally arisen in a different teosinte or maize population and moved between populations via gene flow with maize, an idea suggested by *Ross-Ibarra et al., 2009*, based on allele sharing at a small set of loci. It is worth noting that the total number of shared sweeps is much lower than previously reported (*Tittes et al., 2021*), and only four of fifteen instances of convergence via migration include two or more teosinte and maize populations (*Appendix 4—figure 5*). Despite the lower totals, the evidence still supports this hypothesis. We found relatively few shared sweeps exclusive to teosinte populations (*Figure 5C*), which is what we would expect if maize populations facilitate the movement of beneficial teosinte alleles (via human movement and trade). However, it is important to note that there are fewer sweeps exclusive to teosinte for all modes of convergence, not just via migration, so this alone is not strong evidence. Further, sweeps shared via migration that were exclusive to teosinte were always inferred to be the lowest migration rate ($1\times10^{-3}$) (*Figure 5F*), where those shared by both subspecies or exclusive to maize were primarily inferred to be the highest migration rate ($1\times10^{-1}$). This indicates alleles that are only beneficial in teosinte move between populations more slowly.

Despite the patterns of population structure in both subspecies, there is evidence of gene flow based on $F_4$ tests. In particular, $F_4$ tests that included maize from Crucero Lagunitas were consistently elevated across both subspecies (*Figure 1E* and *Appendix 2—figure 1* and *Appendix 2—table 1*). It was surprising to find that maize from Crucero Lagunitas was featured heavily as a source population, but only for sweeps exclusive to maize (*Figure 5D*). One explanation for this discrepancy is that perhaps movement of this population was common, but the alleles were only beneficial in a domesticated genetic background and have been excluded from teosinte populations except in the case of neutral regions, which could leave a signature in $F_4$ tests but not tests for beneficial mutations. Together, these results suggest that geographically widespread varieties of maize such as Celaya (Crucero Lagunitas) (*Orozco-Ramírez et al., 2017*) may have played a prominent role as a source of and/or transport for beneficial alleles among maize and teosinte populations. However, teosinte populations were often the source of beneficial alleles for sweeps shared between both subspecies (*Figure 5C*). As such, teosinte populations may have also commonly been the source of adaptive variation for both other teosinte and maize populations.

# Materials and methods
## Samples and whole-genome resequencing

We sampled seeds from five populations of *Z. mays* ssp. *parviglumis* and four populations of *Z. mays* ssp. *mays* from plants growing across the species' native range. We additionally included populations of *parviglumis* and maize from Palmar Chico which were previously analyzed and reported in *Chen et al., 2020*, for a total of six and five populations of maize and teosinte (see Appendix 1 for accession IDs and further sample details). All maize and teosinte populations from each named location were

less than 1 km from one another, with the exception of Crucero Lagunitas, which were separated by approximately 18 km.

DNA extraction for teosinte followed (*Chen et al., 2020*). Genomic DNA for landraces was extracted from leaf tissue using the E.Z.N.A. Plant DNA Kit (Omega Biotek), following the manufacturer's instructions. DNA was quantified using a Qubit (Life Technologies) and 1 µg of DNA per individual was fragmented using a bioruptor (Diagenode) with 30 s on/off cycles.

DNA fragments were then prepared for Illumina sequencing. First, DNA fragments were repaired with the End-Repair enzyme mix (New England Biolabs). A deoxyadenosine triphosphate was added at each 3′end with the Klenow fragment (New England Biolabs). Illumina TruSeq adapters (Affymetrix) were then added with the Quick ligase kit (New England Biolabs). Between each enzymatic step, DNA was washed with sera-mags speed beads (Fisher Scientific). The libraries were sequenced to an average coverage of 20–25× PE150 on the Illumina X10 at Novogene (Sacramento, USA).

We additionally grew one individual of *Z. diploperennis* from the UC Davis Botanical Conservatory as an outgroup. DNA for *Z. diploperennis* was extracted and libraries prepared as above, and then sequenced to 60× coverage using PE250 on three lanes of *Haines et al., 2000*, rapid run (UC Davis Genome Center, Davis, USA).

Sequencing reads have been deposited in the NCBI Sequence Read Archive under BioProject ID PRJNA1076902.

## Sequencing and variant identification

All paired-end reads were aligned to version 5 of the maize B73 reference genome (*Hufford et al., 2021*) using bwa-mem (v0.7.17) (*Li, 2013*). Default options were used for mapping except -M to enable marking short hits as secondary, -R for providing the read group, and -K 10,000,000, for processing 10 Mb input in each batch. Sentieon (v201808.01) (*Freed et al., 2017*) was used to process the alignments to remove duplicates (option –algo Dedup) and to calculate various alignment metrics (GC bias, MQ value distribution, mean quality by cycle, and insert size metrics) to ensure proper mapping of the reads.

All downstream analyses were based on genotype likelihoods estimated with *ANGSD* (v0.934) (*Korneliussen et al., 2014*) using the following command line flags and filters:

```
-GL 1
–P 5
-uniqueOnly 1
-remove_bads 1
-only_proper_pairs 1
-trim 0
–C 50
-minMapQ 30
-minQ 30
```

## Genetic diversity

We estimated per base nucleotide diversity ($\pi$) and Tajima's *D* (*D*) in non-overlapping 1 kb, 100 kb, and 1000 kb windows with the *thetaStat* utility from *ANGSD*, though estimates did not substantively differ between window sizes. To estimate $\pi$ and the unfolded site frequency spectra for each population, we polarized alleles as ancestral and derived using short-read sequence data for *Z. luxurians* and *Z. diploperennis* as outgroups. *Z. luxurians* sequence from *Tenaillon et al., 2011*, was downloaded from The NCBI Sequence Read Archive (study SRR088692). We used the alignments from the two species to make minor allele frequency (MAF) files using *ANGSD*. We used the MAF files to construct a table of genotypes found at each locus. Sites with MAF estimates greater than 0.001 were treated as heterozygous. Sites that were homozygous in both species were imputed onto the maize v5 reference and assumed to be the ancestral allele. As there were substantially more called bases in *Z. luxurians* than in *Z. diploperennis*, we also assumed sites that were homozygous in *luxurians* and missing in *diploperennis* were ancestral, but excluded sites that were missing from *luxurians*. Sites that were classified as heterozygous were treated as missing and imputed onto the maize reference as '*N*'.

## Population structure and introgression

We used *ngsadmix* (*Skotte et al., 2013*) to assess population structure within subspecies. To do so we used an SNP-calling procedure in *ANGSD* with the same filters as listed above, along with an SNP p-value cutoff of $1\times10^{-6}$. We looked for evidence of gene flow between subspecies using $F_4$ statistics and *Z*-scores calculated with blocked jackknifing, implemented using Treemix (*Pickrell and Pritchard, 2012*). Trees for $F_4$ tests were always of the form

$$(Maize\_X, Maize\_Y; Teosinte\_focal, Teosinte\_Z),$$

or

$$(Maize\_focal, Maize\_Y; Teosinte\_W, Teosinte\_Z);$$

with each unique combination of populations considered to be the '*focal*' and '*\_X*' positions of the tree. We considered any tree with a *Z*-score greater than or equal to three significant, indicating a departure from the allele frequency covariance expected if population history matched the hypothesized tree. We assessed *Z*-scores separately based on whether the focal population was maize or teosinte, and for trees that include the sympatric pair of the focal population.

## Demographic and inbreeding history

We inferred each population's demography using a single unfolded site frequency spectrum with mushi (v0.2.0) (*DeWitt et al., 2021*). In efforts to reduce overfitting given our modest samples sizes, we increased the regularization penalty parameters to alpha_tv = 1e4, alpha_spline = 1e4, and alpha_ridge = 1e-1.

We assessed HBD in each population using IBDseq (v2.0) (*Browning and Browning, 2013*). We compared empirical results to simulations in msprime (*Kelleher et al., 2016*) using each population's inferred demographic history. We performed 10 replicates of each of these simulations. Replicates were similar across all populations; only one replicate was chosen at random for visual clarity. We estimated recent inbreeding using ngsrelate (*Hanghøj et al., 2019*) with default parameters. Input files were generated using *ANGSD* with the same filters as listed above, along with an SNP p-value cutoff and maf filter of:

```
-SNP_pval 1e-6
-minmaf 0.05
```

respectively.

## Estimating the proportion of fixations due to positive selection, $\alpha$

We modeled the rate of positive selection, $\alpha$ of 0-fold nonsynonymous mutations using the asymptotic extension of the MK test (*Messer and Petrov, 2013*), where $\alpha$ is calculated at each allele frequency bin of the uSFS (from $1/n$ to $(n-1)/n$) (see section 'Genetic diversity' above for further details on uSFS estimation). At each allele frequency bin, each $\alpha$ was calculated as

$$\alpha = 1 - \frac{d_0}{d}\frac{p}{p_0}$$

where $d_0$ and $d$ are the number of derived fixed differences for selected and putatively neutral sites, respectively, and $p_0$ and $p$ are the number of selected and putatively neutral polymorphic sites. We identified 0-fold and 4-fold sites using Python (https://github.com/silastittes/cds_fold, copy archived at *Tittes, 2023a*).

We fit the asymptotic MK extension as a nonlinear Bayesian mixed-model using the R package *brms* (*Bürkner, 2017*; *Bürkner, 2018*).

$$\alpha_{ij} \sim a_j + b_j e^{-c_j x_{ij}}$$

where $\alpha_{ij}$ is the value of $\alpha$ calculated at the $i$th allele frequency bin of the $j$th population and $x_{ij}$ is the corresponding allele frequency bin. The nonlinear brms model was coded as:

```
alpha~
a+b * exp(-c * allele_frequency)
```

where all three free parameters of the asymptotic function ($a$, $b$, and $c$) were treated as random effects of population and nucleotide type (see Appendix 3), and the subspecies was treated as a fixed effect. These effects were coded in brms as:

```
a+b+c
~1+allele_frequency +
(1+allele_frequency | pop)+
(1+allele_frequency | nuc_type)+
ssp
```

## Identifying selective sweeps

We used RAiSD (*Alachiotis and Pavlidis, 2018*) to infer signatures of selective sweeps in each population, including the rangewide samples. Across all populations, we used a MAF threshold of 0.05 and a window size of 24 SNPs. We called SNPs and generated VCF files for each population using the *dovcf* utility from ANGSD, using an SNP-calling p-value cutoff of $1\times10^{-6}$. We used the Python package mop (https://pypi.org/project/mop-bam/) to exclude SNPs that fell in regions with low and excessive coverage and/or poor quality. Here, we required each locus to have at least 70% of individuals with a depth between 5 and 100, and to have a phred scaled quality scores above 30 for base and mapping quality. We additionally used mop to rescale the $\mu_{var}$ sub-statistic in each population based on the proportion of high-quality bases available in each of the RAiSD SNP windows, after which we recalculated the overall $\mu$ statistic as the product of the three sub-statistics (*Alachiotis and Pavlidis, 2018*).

Individual SNP windows in RAiSD are small and spatially autocorrelated with neighboring windows, implying that a sweep signature is distributed over multiple windows. This makes it difficult to compare shared RAiSD outliers across populations, which may have experienced the same sweep, but may differ in how the $\mu$ statistic is distributed across windows. To compare sweep signatures across populations we collected outliers into 'sweep regions' by fitting a cubic spline modeling the adjusted $\mu$ statistic by position along the chromosome using the *smooth.spline* function in R, which uses cross-validation to choose the optimal smoothing parameter for the cubic spline. We compared the empirical cubic spline fit to the those simulated under each population's demography using msprime (*Kelleher et al., 2016*), and selected windows with fitted values greater than 99.9th percentile from the simulations as outlier regions. We merged outlier regions within 50 kb of one another and treated them as a single sweep region. The custom R functions for defining outlier regions are available at https://github.com/silastittes/sweep_regions, copy archived at *Tittes, 2023b*.

For the above steps, we selected the best combination of parameters by conducting a grid search that optimized on the highest proportion of sweeps shared between the two Palmar Chico replicates, averaged over both subspecies. For the grid search we considered RAiSD SNP windows of 10, 24, 50, 100, 200, and 500; outlier quantiles of 0.8, 0.9, 0.95, 0.99, and 0.999; and merged windows of 50K, 100K, 200K, and 500K BPs.

## Assessing the accuracy of sweep inferences and the number of shared sweeps expected by chance

To assess the accuracy of sweep inferences, we used a second random sample of non-overlapping individuals from the two Palmar Chico populations. False positives were assessed based on the number of sweep regions that did not overlap between the two replicate Palmar Chico samples from each subspecies. To account for differences in the total number of sweep regions for each replicate, we averaged the two proportions

$$P = 1 - (\frac{n_S}{n_{P1}} + \frac{n_S}{n_{P2}})/2$$

where $n_S$ is the number of sweeps shared between the replicates, and $n_{P1}$ and $n_{P2}$ were the number of sweeps in the first and second replicates, respectively. In downstream analyses, we used the average value of $P$ for the two subspecies, although it was higher for maize than for teosinte (0.33 and 0.20, respectively).

We evaluated the number of sweeps that we would expect populations to share by chance using a simple statistical test based on the hypergeometric distribution,

$$Pr(x \geq X) = \sum_{x \geq X}^{n_1} \frac{\binom{n_2}{x}\binom{N}{n_1-x}}{\binom{N}{n_1}}$$

where $N$ is the total number of loci tested; $n_1$ and $n_2$ are the number of outlier loci in the first and second populations, respectively; and $x$ is the number of shared outliers between the two populations. The population with the larger number of outliers was always designated at the first population. We accounted for sweep inference accruacy by multiplying the raw values of $N$, $n_1$, $n_2$, and $x$ by the value of $P$ described above.

We corrected p-values for multiple tests using the Benjamini and Yekutieli method implemented with the R function *p.adjust* (*Benjamini and Yekutieli, 2001*).

## Inferring modes of convergent adaptation

For sweep regions that were overlapping in 2–9 of the 11 populations, we used *rdmc* to infer the most likely mode of convergent adaptation (*Lee and Coop, 2017*; *Tittes, 2020*). To ensure a sufficient number of loci were included to estimate decay in covariance across the sweep regions, we added 10% of each sweep region's total length on each of its ends prior to fitting the models. To reduce the computation time, we exclude sites that had an allele frequency less than 1/20 across all populations. As sweep regions differed in size, we subset from their total number of sites to maintain approximately similar densities, with a lower and upper bound of 1K and 100K SNPs, respectively. Sweep regions near the ends of chromosomes for which we could not estimate the number of centiMorgans for were subset to 10K SNPs. To contrast allele frequency covariance in sweep regions to neutral expectations, we first sampled allele frequencies at 100K random loci. When fitting *rdmc*, we assumed the effective population size was 50K for all populations. The recombination rate was approximated for each sweep region as the median interpolated value based on a previously generated genetic map for maize that was lifted on to the v5 reference genome (*Ogut et al., 2015*). The *rdmc* function includes arguments that control the grid of parameter values over which composite likelihoods were computed. These were:

```
n_sites = 200,
num_bins =1000,
sels = 10^seq(-5,-1, length.out=12),
times = c(1e2, 1e3, 1e4, 1e5),
gs = 10^seq(-3,-1, length.out=6),
migs = 10^(seq(-3,-1, length.out=6)),
cholesky = TRUE
```

We compared composite likelihoods over four convergent adaptation modes, 'neutral', 'independent', 'standing', and 'migration'. We assigned each sweep to the mode with the highest log composite likelihood. To assess the overall performance of the method to distinguish between the four modes, we computed differences between the highest composite likelihood and the next highest for each sweep.

All wrangling to prepare input data for statistical analyses was done using *R Development Core Team, 2020*, with appreciable reliance on functions from the tidyverse suite (*Wickham et al., 2019*). Figures were made using ggplot2 (*Wickham, 2016*), patchwork (*Pedersen, 2019*), and cowplot (*Wickham et al., 2019*). Code and instructions for the entirety of the analyses, including Jupyter notebooks for reproducing figures, are available from https://github.com/silastittes/parv_local, copy archived at *Tittes and Seetharam, 2025*.

## Acknowledgements

We would like to thank Andi Kur for providing the corn art, along with Matthew Gibson, Tom Booker, Cathy Rushworth, and members of the Ross-Ibarra lab for feedback and suggestions on early drafts of the manuscript. This work was funded in part by grants from the National Science Foundation (1822330 and 1238014) and Hatch support from the USDA (project CA-D-PLS-2066-H 548). We would also like to acknowledge Felix Andrews for statistical advice, although we did not follow it.

# Additional information

## Competing interests

Jeffrey Ross-Ibarra: Reviewing editor, eLife. The other authors declare that no competing interests exist.

## Funding

| Funder | Grant reference number | Author |
|--------|------------------------|--------|
| National Science Foundation | 1822330 | Jeffrey Ross-Ibarra |
| National Science Foundation | 1238014 | Jeffrey Ross-Ibarra |
| U.S. Department of Agriculture | CA-D-PLS-2066-H 548 | Jeffrey Ross-Ibarra |

The funders had no role in study design, data collection and interpretation, or the decision to submit the work for publication.

## Author contributions

Silas Tittes, Conceptualization, Data curation, Software, Formal analysis, Validation, Investigation, Visualization, Methodology, Writing – original draft, Writing – review and editing; Anne Lorant, Conceptualization, Resources, Data curation; Sean P McGinty, Software, Formal analysis, Visualization; James B Holland, Maud Tenaillon, Conceptualization, Resources, Writing – review and editing; Jose de Jesus Sánchez-González, Resources; Arun Seetharam, Software, Formal analysis, Writing – review and editing; Jeffrey Ross-Ibarra, Conceptualization, Resources, Data curation, Software, Formal analysis, Supervision, Funding acquisition, Validation, Investigation, Visualization, Methodology, Writing – original draft, Writing – review and editing

## Author ORCIDs

Silas Tittes ⓘ https://orcid.org/0000-0003-4697-7434
Jeffrey Ross-Ibarra ⓘ https://orcid.org/0000-0003-1656-4954

Reviewer #1 (Public review): https://doi.org/10.7554/eLife.92405.3.sa1
Author response https://doi.org/10.7554/eLife.92405.3.sa2

# Additional files

## Supplementary files

MDAR checklist

## Data availability

Sequencing reads have been deposited in the NCBI Sequence Read Archive under BioProject ID PRJNA1076902. All figures, summary statistics, and tables can be generated using the pipeline and notebooks available at GitHub, copy archived at *Tittes and Seetharam, 2025*.

The following dataset was generated:

| Author(s) | Year | Dataset title | Dataset URL | Database and Identifier |
|-----------|------|---------------|-------------|-------------------------|
| Tittes S | 2024 | The population genetics of convergent adaptation in maize and teosinte | https://www.ncbi.nlm.nih.gov/bioproject/PRJNA1076902 | NCBI BioProject, PRJNA1076902 |

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

# Appendix 1

## Population sampling locations

**Appendix 1—table 1.** Population sampling location information.

| Population | Subspecies | Sample size | Latitude | Longitude | Elevation (m) | Accession ID |
|---|---|---|---|---|---|---|
| Crucero Lagunitas | Maize | 10 | 16.98 | −99.28 | 201 | 2373-GRO-294 |
| Amatlán de Cañas | Maize | 10 | 20.82 | −104.41 | 760 | 5054-NAY-310 |
| Los Guajes | Maize | 10 | 19.23 | −100.49 | 985 | TC-300 |
| San Lorenzo | Maize | 10 | 19.94 | −103.99 | 982 | RMM-15 |
| Palmar Chico | Maize | 55 | 18.64 | −100.35 | 1008 | JSG-RMM-LCL-529 |
| Crucero Lagunitas | Teosinte | 10 | 16.85 | −99.06 | 590 | JSG-RMM-LCL-487 |
| Amatlán de Cañas | Teosinte | 10 | 20.82 | −104.41 | 880 | JSG-JRP-ERG-543 |
| El Rodeo | Teosinte | 10 | 16.35 | −97.02 | 982 | JSG-RMM-LCL-486 |
| Los Guajes | Teosinte | 10 | 19.23 | −100.49 | 851 | JSG Y RMM-454 |
| San Lorenzo | Teosinte | 10 | 19.94 | −103.99 | 982 | RMM-13 |
| Palmar Chico | Teosinte | 50 | 18.64 | −100.35 | 983 | JSG-RMM-LCL-528 |

## Appendix 2

## Further assessment of $F_4$ statistic inferences

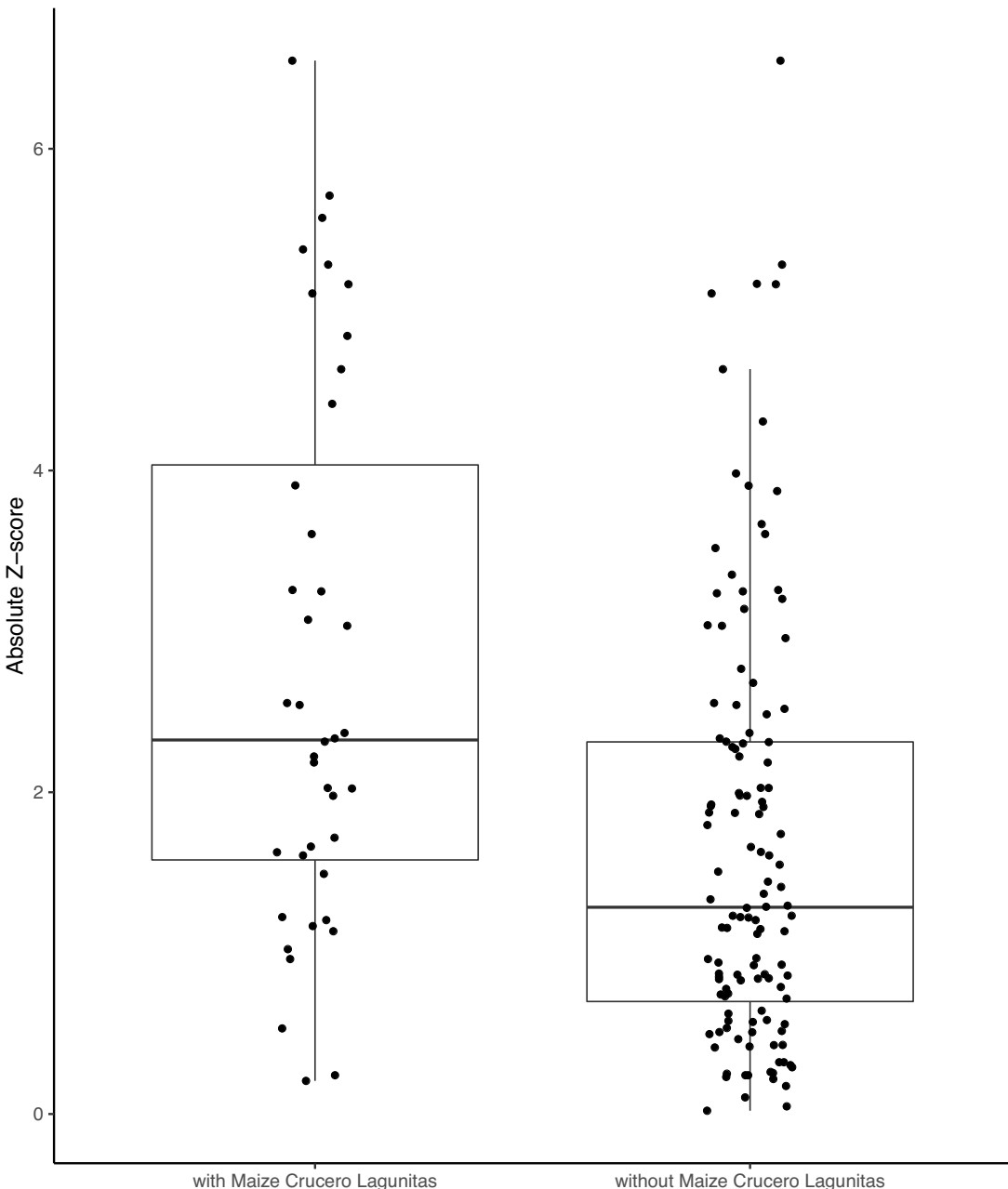

**Appendix 2—figure 1.** $F_4$ tests including the maize Crucero Lagunitas population are significantly elevated compared to those without.

**Appendix 2—table 1.** Significant $F_4$ tests.
Each row of the table reports the number of significant $F_4$ tests that occurred with a given focal and secondary population, where the two other tip positions were filled with each of the remaining populations for each subspecies. Rows that are left blank in the secondary column are used to report the total number of significant trees for a given focal population.

*Appendix 2—table 1 Continued on next page*

| Focal population | Secondary population | Count |
|---|---|---|
| Maize Amatlan de Canas | | 5 |
| Maize Amatlan de Canas | Maize Crucero Lagunitas | 5 |
| Maize Amatlan de Canas | Teosinte Amatlan de Canas | 3 |
| Maize Amatlan de Canas | Teosinte El Rodeo | 2 |
| Maize Amatlan de Canas | Teosinte Palmar Chico | 2 |
| Maize Amatlan de Canas | Teosinte San Lorenzo | 2 |
| Maize Amatlan de Canas | Teosinte Los Guajes | 1 |
| Maize Crucero Lagunitas | | 15 |
| Maize Crucero Lagunitas | Teosinte Amatlan de Canas | 9 |
| Maize Crucero Lagunitas | Teosinte Crucero Lagunitas | 6 |
| Maize Crucero Lagunitas | Teosinte El Rodeo | 6 |
| Maize Crucero Lagunitas | Maize Palmar Chico | 5 |
| Maize Crucero Lagunitas | Maize Los Guajes | 4 |
| Maize Crucero Lagunitas | Teosinte San Lorenzo | 4 |
| Maize Crucero Lagunitas | Maize Amatlan de Canas | 3 |
| Maize Crucero Lagunitas | Maize San Lorenzo | 3 |
| Maize Crucero Lagunitas | Teosinte Palmar Chico | 3 |
| Maize Crucero Lagunitas | Teosinte Los Guajes | 2 |
| Maize Los Guajes | | 6 |
| Maize Los Guajes | Teosinte Amatlan de Canas | 4 |
| Maize Los Guajes | Maize Crucero Lagunitas | 3 |
| Maize Los Guajes | Teosinte Crucero Lagunitas | 3 |
| Maize Los Guajes | Maize San Lorenzo | 2 |
| Maize Los Guajes | Teosinte Palmar Chico | 2 |
| Maize Los Guajes | Maize Palmar Chico | 1 |
| Maize Los Guajes | Teosinte El Rodeo | 1 |
| Maize Los Guajes | Teosinte Los Guajes | 1 |
| Maize Los Guajes | Teosinte San Lorenzo | 1 |
| Maize Palmar Chico | | 9 |
| Maize Palmar Chico | Teosinte Amatlan de Canas | 7 |
| Maize Palmar Chico | Maize Crucero Lagunitas | 5 |
| Maize Palmar Chico | Teosinte Palmar Chico | 4 |
| Maize Palmar Chico | Teosinte El Rodeo | 3 |
| Maize Palmar Chico | Maize Los Guajes | 2 |
| Maize Palmar Chico | Maize San Lorenzo | 2 |
| Maize Palmar Chico | Teosinte San Lorenzo | 2 |
| Maize Palmar Chico | Teosinte Crucero Lagunitas | 1 |
| Maize Palmar Chico | Teosinte Los Guajes | 1 |
| Maize San Lorenzo | | 6 |
| Maize San Lorenzo | Teosinte Amatlan de Canas | 4 |
| Maize San Lorenzo | Maize Crucero Lagunitas | 3 |
| Maize San Lorenzo | Maize Los Guajes | 2 |

| Focal population | Secondary population | Count |
|---|---|---|
| Maize San Lorenzo | Teosinte Crucero Lagunitas | 2 |
| Maize San Lorenzo | Teosinte El Rodeo | 2 |
| Maize San Lorenzo | Teosinte Palmar Chico | 2 |
| Maize San Lorenzo | Maize Palmar Chico | 1 |
| Maize San Lorenzo | Teosinte Los Guajes | 1 |
| Maize San Lorenzo | Teosinte San Lorenzo | 1 |
| Teosinte Amatlan de Canas | | 11 |
| Teosinte Amatlan de Canas | Maize Crucero Lagunitas | 9 |
| Teosinte Amatlan de Canas | Maize Palmar Chico | 4 |
| Teosinte Amatlan de Canas | Maize Amatlan de Canas | 3 |
| Teosinte Amatlan de Canas | Maize Los Guajes | 3 |
| Teosinte Amatlan de Canas | Maize San Lorenzo | 3 |
| Teosinte Amatlan de Canas | Teosinte Los Guajes | 3 |
| Teosinte Amatlan de Canas | Teosinte Palmar Chico | 3 |
| Teosinte Amatlan de Canas | Teosinte San Lorenzo | 3 |
| Teosinte Amatlan de Canas | Teosinte Crucero Lagunitas | 1 |
| Teosinte Amatlan de Canas | Teosinte El Rodeo | 1 |
| Teosinte Crucero Lagunitas | | 9 |
| Teosinte Crucero Lagunitas | Maize Crucero Lagunitas | 6 |
| Teosinte Crucero Lagunitas | Maize Los Guajes | 5 |
| Teosinte Crucero Lagunitas | Teosinte Amatlan de Canas | 4 |
| Teosinte Crucero Lagunitas | Teosinte El Rodeo | 4 |
| Teosinte Crucero Lagunitas | Maize Palmar Chico | 3 |
| Teosinte Crucero Lagunitas | Maize San Lorenzo | 3 |
| Teosinte Crucero Lagunitas | Maize Amatlan de Canas | 1 |
| Teosinte Crucero Lagunitas | Teosinte Palmar Chico | 1 |
| Teosinte Los Guajes | Maize Crucero Lagunitas | 3 |
| Teosinte Los Guajes | Teosinte Amatlan de Canas | 3 |
| Teosinte Los Guajes | | 3 |
| Teosinte Los Guajes | Maize Los Guajes | 1 |
| Teosinte Los Guajes | Maize Palmar Chico | 1 |
| Teosinte Los Guajes | Maize San Lorenzo | 1 |
| Teosinte Palmar Chico | | 8 |
| Teosinte Palmar Chico | Maize Crucero Lagunitas | 5 |
| Teosinte Palmar Chico | Teosinte Amatlan de Canas | 5 |
| Teosinte Palmar Chico | Maize Palmar Chico | 4 |
| Teosinte Palmar Chico | Maize Los Guajes | 3 |
| Teosinte Palmar Chico | Maize San Lorenzo | 3 |
| Teosinte Palmar Chico | Teosinte El Rodeo | 2 |
| Teosinte Palmar Chico | Maize Amatlan de Canas | 1 |
| Teosinte Palmar Chico | Teosinte Crucero Lagunitas | 1 |
| Teosinte San Lorenzo | Maize Crucero Lagunitas | 5 |

| Focal population | Secondary population | Count |
|---|---|---|
| Teosinte San Lorenzo | | 5 |
| Teosinte San Lorenzo | Teosinte Amatlan de Canas | 3 |
| Teosinte San Lorenzo | Maize Palmar Chico | 2 |
| Teosinte San Lorenzo | Teosinte El Rodeo | 2 |
| Teosinte San Lorenzo | Maize Amatlan de Canas | 1 |
| Teosinte San Lorenzo | Maize Los Guajes | 1 |
| Teosinte San Lorenzo | Maize San Lorenzo | 1 |

# Appendix 3

## Additional analyses related to $\alpha$

### Predicting $\alpha$ by mutation type

Estimates of $\alpha$ may be effected by differences in the mutation rates of different nucleotides and genomic regions. GC-biased gene conversion has been shown to reduce $\alpha$ by making it harder to purge slightly deleterious alleles (**Hämälä and Tiffin, 2020**). Likewise, the higher mutation rates observed at methylated cytosine bases increase the rate of $C \rightarrow T$ mutations (**Ossowski et al., 2010**), which is another mechanism that could result in variation in $\alpha$ by changing the ability to purging deleterious alleles, or by changing the probability of fixation of new adaptive mutations.

To study this, we used the same approach as **Hämälä and Tiffin, 2020**, where we separated the site frequency spectra based on mutation types according to whether the ancestral and derived nucleotides had a single (weak) or double (strong) hydrogen bond between the DNA strands. As such, we studied three mutations types: $A/T \rightarrow G/C$ mutations (WS), $G/C \rightarrow A/T$ (SW), and $C/G \rightarrow G/C$ or $A/T \rightarrow T/A$ (SS_WW).

Unlike patterns found in *Arabidopsis* (**Hämälä and Tiffin, 2020**), $\alpha$ was highest for WS mutations, although there was considerable overlap between the credible intervals for all mutation types.

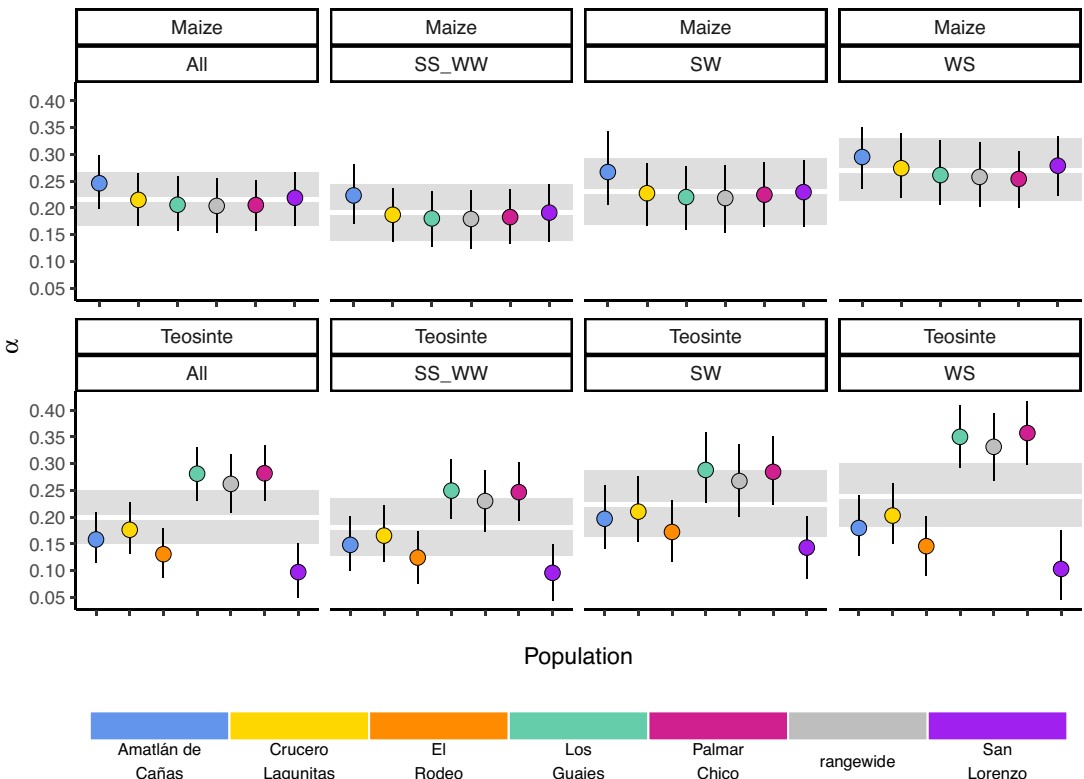

**Appendix 3—figure 1.** Predicted values of $\alpha$ across mutation types. Gray bands for each mutation type show the 95% credible intervals averaged over each population.

### Amount of unique fixations across teosinte populations

**Appendix 3—table 1.** Proportion and count of fixed differences at 0-fold sites across teosinte populations.

| Population | Proportion | Total |
|---|---|---|
| Amatlan de Canas | 0.12238 | 1195 |
| Crucero Lagunitas | 0.15619 | 1880 |

*Appendix 3—table 1 Continued on next page*

*Appendix 3—table 1 Continued*

| Population | Proportion | Total |
|---|---|---|
| El Rodeo | 0.23233 | 2983 |
| Los Guajes | 0.03025 | 170 |
| San Lorenzo | 0.33796 | 6050 |
| Palmar Chico | 0.03580 | 184 |

## Appendix 4

### Further assessment of sweep inferences and precision

We found that only 67% and 80% of sweeps were shared between the two random subsamples for maize and teosinte populations from Palmar Chico, respectively (see Results). While our updated method to identifying sweeps improved precision over our previous one (which shared 40% and 50%; *Tittes et al., 2021*), the low precision still warrants further exploration.

One explanation for the low sweep precision could be substructure within the Palmar Chico populations, leading to replicates with slightly different histories. This could create unequal power to detect sweeps if, for example, the subpopulations had different progress toward fixation of the beneficial allele. However, this explanation is unlikely as individuals were randomly assigned to subpopulations, so any substructure is likely to be distributed evenly between the two samples. This is further supported from the two samples showing relatively short branch lengths in the population phylogeny (*Figure 1*). The branch lengths separating the two subsamples (cophenetic distance) were 0.00835 and 0.00962 for maize and teosinte respectively, compared to the within subspecies means of 0.0170 and 0.0559.

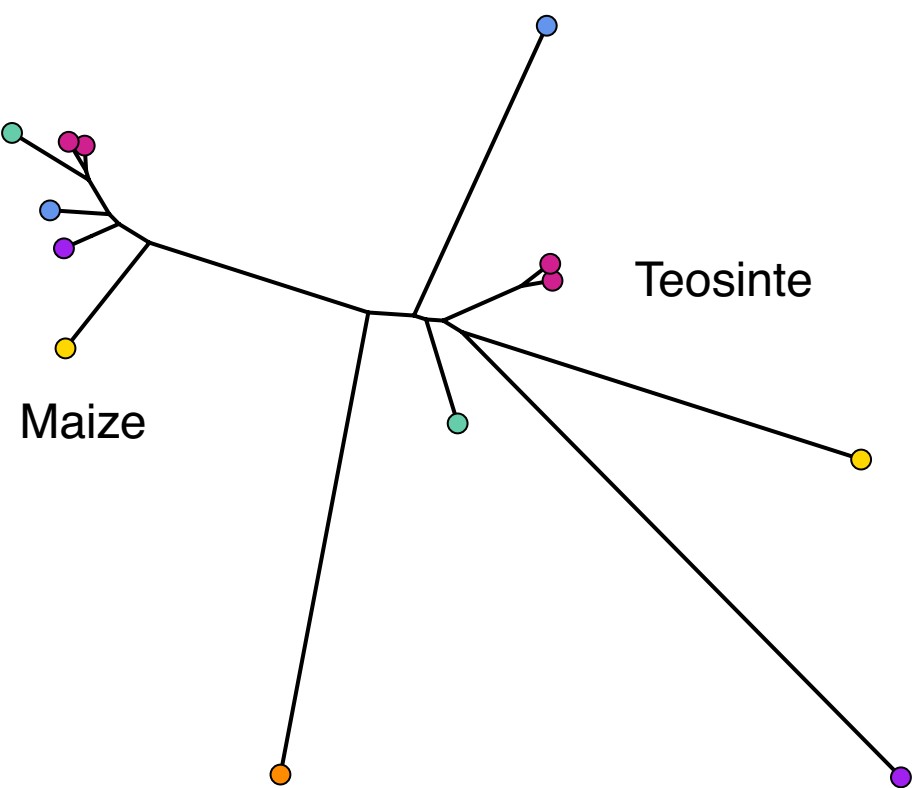

**Appendix 4—figure 1.** Treemix phylogeny including both subsamples of Palmar Chico.

Another potential explanation for lowered sweep sharing between replicates is that sweeps vary in their detectability based on their characteristics. Namely, sweeps that were weakly selected, incomplete, and/or ones that started at a high initial frequency prior to the onset of selection (soft sweeps) may vary in their detectability using the methods we employed. We conducted a simulation experiment to better understand the potential causes of the low shared proportion, and to measure performance to detect different kinds of sweeps more generally. We used discoal (*Kern and Schrider, 2016*) to simulate sweeps in a 400 kb region using the average genome-wide maize mutation and recombination rates under the inferred demographic history of the maize population from Palmar Chico (*Figure 2*). We simulated four distinct scenarios: classical hard sweeps, where selection acts to fix an an adaptive mutation; soft sweeps, where selection is initiated after the adaptive allele reaches a specified frequency; and incomplete sweeps, where a hard sweep simulation is stopped at a specified frequency, and neutral simulations without selection. For soft sweeps, we varied the initial

by drawing from a beta distribution with shape parameters 1 and 20. Incomplete sweeps finished when the adaptive allele reached a frequency of 0.5. For all three types of sweeps, we also varied the strength of selection using the parameter $\alpha = 4N0s$ to be 10, 50, or 100, where $N0$ is the present-day effective population size and $s$ is the selection coefficient. In addition to matching demography and other parameters, we used the same sampling scheme, simulation 50 individuals, than randomly choosing two non-overlapping subsets of 10 individuals (https://github.com/silastittes/ms_sub, copy archived at *Tittes, 2022*). From the simulations we assessed the True/False Positive/Negative Rates for each combination of sweep type and strength of selection ($\alpha$), as well as the distribution of base pair overlap between sweep regions inferred in the two random subsamples. The same sweep inference methods and parameters were used for these simulations and the empirical samples (see Materials and methods). Overall, we found that sweep characteristics we explore indeed impacted our overall power to detect them, and the about of overlap between the sweep regions. Namely, weakly selected sweeps had consistently lower true positive rates (*Appendix 4—table 1*).

**Appendix 4—table 1.** Performance to detect simulated hard, soft, and incomplete sweeps under varying strengths of selection under the maize Palmar Chico population demography.
TPR, TNR, FNR, and FPR stand for true positive, true negative, false negative, and false positive rates, respectively.

| $4N_e s$ | Simulation type | TPR | TNR | FNR | FPR |
|---|---|---|---|---|---|
| 10 | Hard | 0.14 | 0.67 | 0.86 | 0.33 |
| 50 | Hard | 0.92 | 0.75 | 0.08 | 0.25 |
| 100 | Hard | 0.99 | 0.84 | 0.01 | 0.16 |
| 10 | Incomplete | 0.04 | 0.67 | 0.96 | 0.33 |
| 50 | Incomplete | 0.06 | 0.67 | 0.94 | 0.33 |
| 100 | Incomplete | 0.05 | 0.63 | 0.95 | 0.37 |
| 10 | Soft | 0.13 | 0.71 | 0.87 | 0.29 |
| 50 | Soft | 0.75 | 0.74 | 0.25 | 0.26 |
| 100 | Soft | 0.81 | 0.76 | 0.19 | 0.24 |

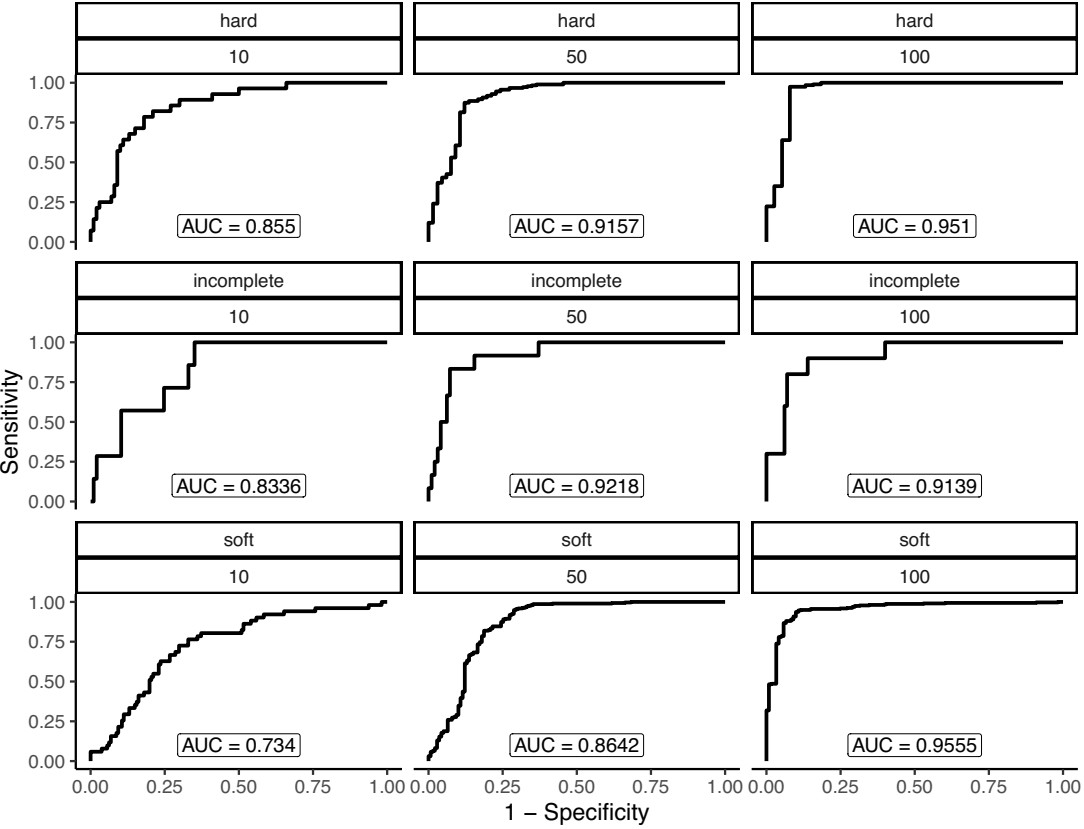

**Appendix 4—figure 2.** Performance to detect simulated hard, soft, and incomplete sweeps under varying strengths of selection under the maize Palmar Chico population demography. Each panel shows a combinations of sweep type (hard, soft, or incomplete) and strength of selection ($\alpha=4N_e s$=10, 50, or 100).

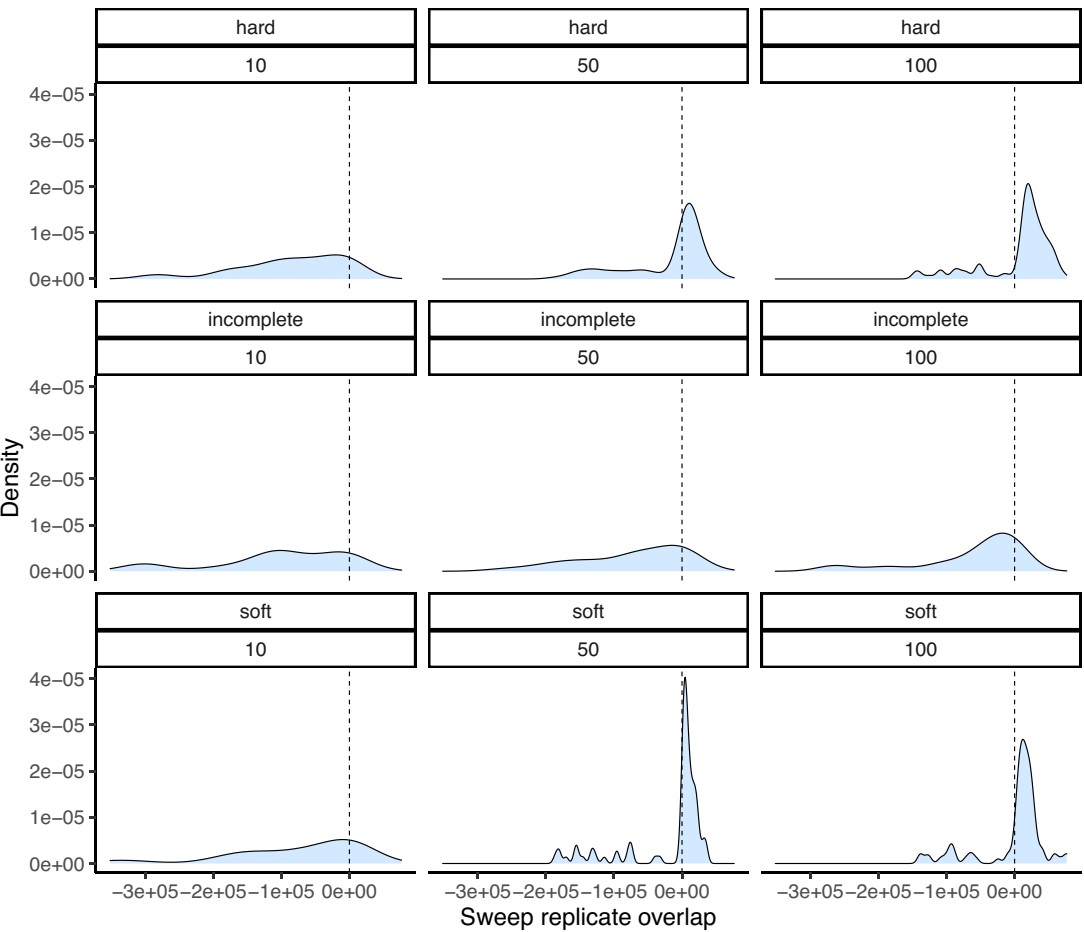

**Appendix 4—figure 3.** Degree of overlap between simulated sweep regions taken from two down-sampled replicates under the maize Palmar Chico population demography. Positive values show the amount of overlap in base pairs between sweep regions, while negative values represent the space between them. Panel structure follows that of *Appendix 4—figure 2*.

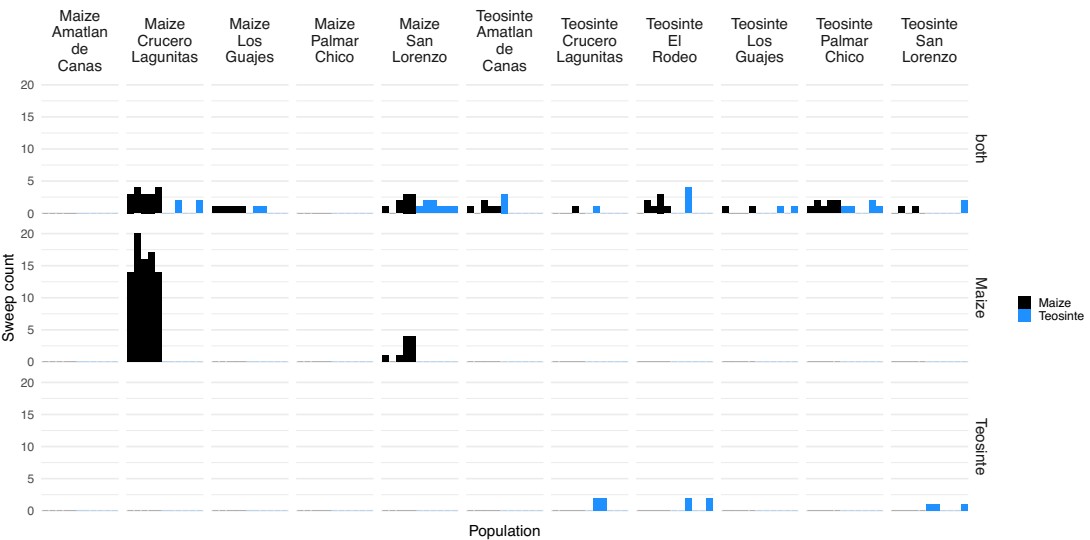

**Appendix 4—figure 4.** Frequency of each population as the mutation source for sweeps shared via migration. The order of populations along the x axis matches that of the source populations labeled for each strip along the top.

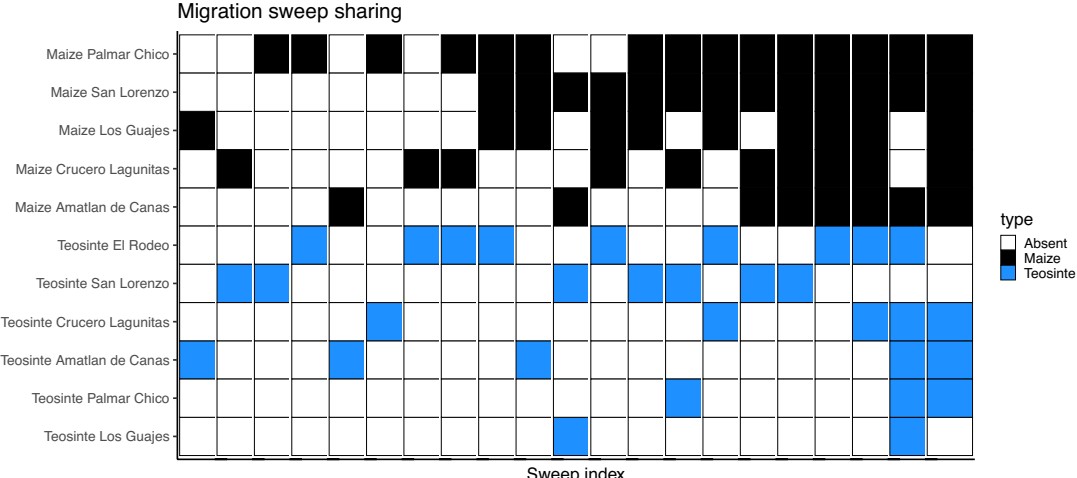

**Appendix 4—figure 5.** Inferred sweeps shared between subspecies via migration. The x axis is sorted by the number of populations each sweep was found in. Populations are sorted along the y axis first by subspecies then by their number of sweeps.

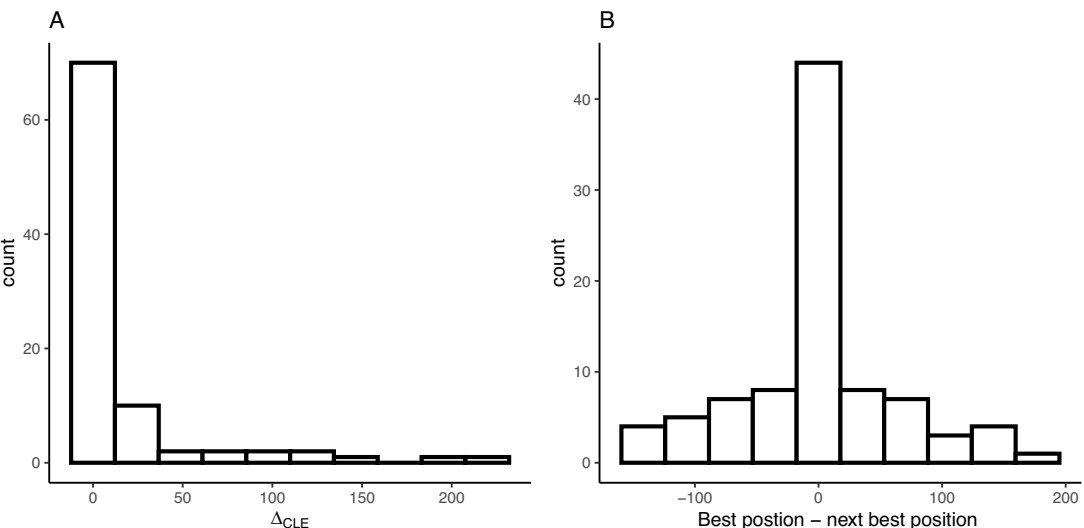

**Appendix 4—figure 6.** Variation in composite likelihoods within sweep regions. (**A**) Distribution of differences between highest and next highest composite likelihoods within each sweep region. (**B**) Distribution of differences between highest and next highest composite likelihoods candidate beneficial mutation positions within each sweep region.

