## [Editor Report · eLife Assessment]

This **useful** study examines patterns of diversity and divergence in two closely related sub-species of Zea mays, patterns that have bearings on local adaptation in maize and teosinte at intermediate geographic scales. The authors suggest that convergent evolution has been facilitated by both standing variation and gene flow, with independent selective sweeps in the two species. While the data themselves are **solid**, there are limitations concerning population sampling, false positive rates in sweep detection and integration of phenotypic data, which make it difficult to draw definitive conclusions. The work should in principle be of broad interest to colleagues studying the relationship between domesticated species and their progenitors, as well as those studying instances of parallel evolution.

---

## [Referee Report · Reviewer #1 (Public review)]

Summary:

This paper examines patterns of diversity and divergence in two closely related sub-species of Zea mays. While the data are interesting and the authors have tried to exclude multiple confounding factors, many patterns cannot clearly be ascribed to one cause or another.

Strengths:

The paper presents interesting data from sets of sympatric populations of the two sub-species, maize and teosinte. This sampling offers unique insights into the diversity and divergence between the two, as well as the geographic structure of each. Many analyses and simulations to check analyses have been carried out.

Weaknesses:

The strength of conclusions that can be drawn from the analyses was low, partly because there are many strange patterns. The authors have done a good job of adding caveats, but clearly, these species do not meet many assumptions of our methods

---

## [Author Response]

The following is the authors’ response to the current reviews.

**Public Reviews:**

**Reviewer #1 (Public review):**
Summary:This paper examines patterns of diversity and divergence in two closely related sub-species of Zea mays. While the data are interesting and the authors have tried to exclude multiple confounding factors, many patterns cannot clearly be ascribed to one cause or another.Strengths:The paper presents interesting data from sets of sympatric populations of the two sub-species, maize and teosinte. This sampling offers unique insights into the diversity and divergence between the two, as well as the geographic structure of each. Many analyses and simulations to check analyses have been carried out.Weaknesses:The strength of conclusions that can be drawn from the analyses was low, partly because there are many strange patterns. The authors have done a good job of adding caveats, but clearly, these species do not meet many assumptions of our methods.

Thank you for the comments. We appreciate the multiple rounds of revision the manuscript has undergone and the work has improved as a consequence. Overall we disagree that the patterns are strange, and have made considerable efforts to explain in the text and in our responses why the patterns make sense based on what we know about the history of Zeamays from previous research. We agree that currently available methods are not capable of answering all questions we propose adequately. This reflects both limitations with the available data for these populations (i.e. phenotypes and spatially explicit sampling), and limitations in available methods tailored to the questions at hand (spatially explicit inference of the range over which an allele is adaptive). We have made considerable effort to point out the places where our inferences are likely to have low accuracy or limited resolution. These limitations are in many ways inherent to all inferential based science and should not be considered a weak point specific to this work, nor do they take away from the fundamental conclusions, which have changed quantitatively but not qualitatively over the course of peer review.

**Recommendations for the authors:**
Reviewer #1 (Recommendations for the authors):-The manuscript should say something about the fact that range-wide PSMC does not show a decline.

We did not use PSMC methods but instead mushi as described in the methods. On line 356 we described how the lower sample size and strong regularization are the most likely explanations for the lack of a population size decline in the rangewide samples.

- The manuscript should explain how rdmc was run and what "overlapping" means.

We described how sweep intervals were inferred starting on line 823 (Methods subsection “Identifying Selective Sweeps”). Sweep regions were defined as the outermost coordinates from all populations that shared any overlap in their respectively defined sweep intervals. The details of how we ran rdmc, including all of the parameters, is described starting on line 895 (methods subsection “Inferring modes of convergent adaptation”).

- Figure 4: "Negative log10" is messed up

Thank you. This has been fixed for the Version Of Record.

- Line 318: "accruacy"

Thank you. We have edited this typo for the Version Of Record.

- New Table S3: why don't the proportions add to 1?

These values represent what proportion of fixed differences at 0 fold sites are unique to each population. The denominator is the total number of fixed differences for each population separately, so each proportion is distinct for each population and thus should not sum to one across them. The table caption has been reworded in efforts to clarify for the Version Of Record.

The following is the authors’ response to the original reviews.

**Public Reviews:**

**Reviewer #1 (Public Review):**
Summary:This paper examines patterns of diversity and divergence in two closely related sub-species of Zea mays. While the patterns are interesting, the strength of evidence in support of the conclusions drawn from these patterns is weak overall. Most of the main conclusions are not supported by convincing analyses.Strengths:The paper presents interesting data from sets of sympatric populations of the two sub-species, maize and teosinte. This sampling offers unique insights into the diversity and divergence between the two, as well as the geographic structure of each.Weaknesses:There were issues with many parts of the paper, especially with the strength of conclusions that can be drawn from the analyses. I list the major issues in the order in which they appear in the paper.(1) Gene flow and demography.The f4 tests of introgression (Figure 1E) are not independent of one another. So how should we interpret these: as gene flow everywhere, or just one event in an ancestral population? More importantly, almost all the significant points involve one population (Crucero Lagunitas), which suggests that the results do not simply represent gene flow between the sub-species. There was also no signal of increased migration between sympatric pairs of populations. Overall, the evidence for gene flow presented here is not convincing. Can some kind of supporting evidence be presented?

We agree that the standard approach to f4 tests that we employed here is not without limitations, namely, that the tests are conducted independently, while the true evolutionary history is not. While a joint demographic inference across all populations would be useful, it did not seem tractable to perform over all of our populations with currently available methods, given the number of populations being analyzed, nor does it directly address the question of interest. Our purpose for including the f4 was testing if there was more gene flow between sympatric pairs than in other comparisons we have made that point more clear in the text near line 174. As described in the text, the distribution of Z scores is generated by pairing focal populations with all other non-focal populations across both subspecies, which means the gene flow signal of interest is marginalized over the effects of gene flow in the other non-focal populations. This is not nearly as rich as inferring the full history, but it gives us some sense of the average amount of gene flow experienced between populations and allows us to address one of our primary questions of interest when conceiving this paper － do sympatric pairs show more geneflow than other pairs? We agree with the reviewer that that answer is largely no, and the writing reflects this.

Overall, we think both points mentioned by the reviewer here; finding that most but not all tests involved Crucero Lagunitas maize, and that sympatric pairs don’t show higher gene flow; nicely contributes to the overall theme in the paper － the history of both subspecies is idiosyncratic and impacted by humans in ways that do not reflect geographic proximity that we did not anticipate (see expectations near line 110). We have emphasized the connection between f4 tests and the revised rdmc results near line 653.

The paper also estimates demographic histories (changes in effective population sizes) for each population, and each sub-species together. The text (lines 191-194) says that "all histories estimated a bottleneck that started approximately 10 thousand generations ago" but I do not see this. Figure 2C (not 2E, as cited in the text) shows that teosinte had declines in all populations 10,000 generations ago, but some of these declines were very minimal. Maize has a similar pattern that started more recently, but the overall species history shows no change in effective size at all. There's not a lot of signal in these figures overall.I am also curious: how does the demographic model inferred by mushi address inbreeding and homozygosity by descent (lines 197-202)? In other words, why does a change in Ne necessarily affect inbreeding, especially when all effective population sizes are above 10,000?

All maize populations show a decline beginning 10,000 generations ago. The smallest decline for maize is from 100,000 to 30,000. All teosinte populations show a reduction in population size. The smallest of these drops more than 70% from around 300,000 to 100,000. Three of the teosinte populations showed a reduction in population size from ~10^5 to ~10^3, which is well below 10,000. Thus all populations show declines.

These large reductions should lead to inbreeding and increased homozygosity by descent. Mushi does not specifically model these features of the data, yet as we show, simulations under the model estimated by Mushi matched the true HBD levels fairly well (Figure 2D).

The rangewide sample does not show declines, likely because there is enough isolation between populations that the reduction in variation at any given locus is not shared, and is maintained in the populations that did not experience the population decline.

(2) Proportion of adaptive mutations.The paper estimates alpha, the proportion of nonsynonymous substitutions fixed by positive selection, using two different sampling schemes for polymorphism. One uses range-wide polymorphism data and one uses each of the single populations. Because the estimates using these two approaches are similar, the authors conclude that there is little local adaptation. However, this conclusion is not justified.There is little information as to how the McDonald-Kreitman test is carried out, but it appears that polymorphism within either teosinte or maize (using either sampling scheme) is compared to fixed differences with an outgroup. These species might be Z. luxurians or Z. diploperennis, as both are mentioned as outgroups. Regardless of which is used, this sampling means that almost all the fixed differences in the MK test will be along the ancestral branch leading to the ancestor of maize or teosinte, and on the branch leading to the outgroup. Therefore, it should not be surprising that alpha does not change based on the sampling scheme, as this should barely change the number of fixed differences (no numbers are reported).The lack of differences in results has little to do with range-wide vs restricted adaptation, and much more to do with how MK tests are constructed. Should we expect an excess of fixed amino acid differences on very short internal branches of each sub-species tree? It makes sense that there is more variation in alpha in teosinte than maize, as these branches are longer, but they all seem quite short (it is hard to know precisely, as no Fst values or similar are reported).

The section “Genetic Diversity” in the methods provides details about how luxurians and diploperennis were used as outgroups. The section “Estimating the Rate of Positive Selection, α”, in the methods includes the definition of α and full joint non-linear regression equation and the software used to estimate it (brms), and the relevant citations crediting the authors of the original method. However, some of the relevant information about the SFS construction is provided in the previous section entitled, “Genetic Diversity”. We added reference to this in results near line 800.

While we appreciate the concern that “almost all the fixed differences in the MK test will be along the ancestral branch leading to the ancestor of maize or teosinte”, this is only a problem if there aren’t enough fixed differences that are unshared between populations. This is more of a concern for maize than teosinte, which we make clear as a caveat in the manuscript in several places already. The fact that there is variation in alpha among teosinte populations is evidence that these counts do differ among pops. As we can see in the population trees in Figure 1, there is a considerable amount of terminal branch length for all the populations. Indeed if we look at the number of fixed differences at 0 fold sites across populations:

The variation in the number of fixed differences, particularly across teosinte means that a large number cannot be shared between populations. We can estimate the fixed differences unique to each subpopulation (and total count) demonstrating that, in general, there are a large number of substitutions unique to each population. This is good evidence the rangewide estimates do not reflect a lack of variation within populations, at least not for teosinte. This is now included in the supplement (Table S3).

Finally, we note that the branches leading to outgroups are likely not substantially longer than those among populations. Given our estimates of Ne, the coalescent within maize and teosinte should be relatively deep (with Ne of 30K it should be ~120K years). The divergence time between Zea mays and these outgroup taxa has been estimated at ~150K years (Chen et al. 2022). This is now mentioned in the text on line 407.

We have added a caveat about the reviewers concern for the non-independence of fixed difference for maize near line 386.

(3) Shared and private sweeps.In order to make biological inferences from the number of shared and private sweeps, there are a number of issues that must be addressed.One issue is false negatives and false positives. If sweeps occur but are missed, then they will appear to be less shared than they really are. Table S3 reports very high false negative rates across much of the parameter space considered, but is not mentioned in the main text. How can we make strong conclusions about the scale of local adaptation given this? Conversely, while there is information about the false positive rate provided, this information doesn't tell us whether it's higher for population-specific events. It certainly seems likely that it would be. In either case, we should be cautious saying that some sweeps are "locally restricted" if they can be missed more than 85% of the time in a second population or falsely identified more than 25% of the time in a single population.

The reviewer brings up a worthwhile point. The simulation results indeed call into question how many of the sweeps we claim are exclusive to one population actually are. This caveat is already made, but we now make clearer the reviewer’s concern regarding the high false negative rate (near line 299). However, if anything this suggests sweeps are shared even more often than what is reported. One of the major takeaways from the paper is that convergent adaptation is more common than we expected. The most interesting part about the unique sweeps is the comparison between maize and teosinte. While the true proportions may vary, the relatively higher proportion of sweeps exclusive to one population in teosinte compared to maize is unlikely to be affected by false negatives, since the accuracy to identify sweeps pretty similar across subspecies (though perhaps with some exceptions for the populations with stronger bottlenecks). Further, these criticisms are specific to the raisd results. All sweeps shared across multiple populations were analyzed using rdmc. After adjustments made to the number of proposed sites for selection (see response below), there is good agreement between the raisd and rdmc results － the regions we proposed as selective sweeps with raisd all show evidence convergence using rdmc. Recall too that rdmc uses a quite different approach to inference － all populations are used jointly, labelling those that did and did not experience the sweep. If sweeps were present in populations that were labeled as neutral (or vice versa), this would weaken the power to infer selection at the locus. Much of the parameter space we explored is for quite weak selection, and the simulated analysis shows we are likely to miss those instances, often entirely. For strong sweeps, however, our simulations show we have appreciable accuracy.

Together, there is reason to be optimistic about our detection of strong shared sweeps and that the main conclusions we make are sound.

Finally, we note that we are unaware of any other empirical study that has performed similar estimates of the accuracy of the sweep calling in their data (as opposed to using simulations). We thus see these analyses as a significant contribution towards transparency that is completely lacking from most papers.

A second, opposite, issue is shared ancestral events. Maize populations are much more closely related than teosinte (Figure 2B). Because of this, a single, completed sweep in the ancestor of all populations could much more readily show a signal in multiple descendant populations. This is consistent with the data showing more shared events (and possibly more events overall). There also appear to be some very closely (phylogenetically) related teosinte populations. What if there's selection in their shared ancestor? For instance, Los Guajes and Palmar Chico are the two most closely related populations of teosinte and have the fewest unique sweeps (Figure 4B). How do these kinds of ancestrally shared selective events fit into the framework here?

The reviewer brings up another interesting point and one that likely impacts some of our results.

As the reviewer describes, this is an issue that is of more concern to the more closely related populations and is less likely to explain results across the subspecies. We have added this as a caveat (near line 456). As is clear in the writing, sharing across subspecies is our primary interest for the rdmc results.

These analyses of shared sweeps are followed by an analysis of sweeps shared by sympatric pairs of teosinte and maize. Because there are not more events shared by these pairs than expected, the paper concludes that geography and local environment are not important. But wouldn't it be better to test for shared sweeps according to the geographic proximity of populations of the same sub-species? A comparison of the two sub-species does not directly address the scale of adaptation of one organism to its environment, and therefore it is hard to know what to conclude from this analysis.

We did not intend to conclude that local adaptation is not important. Especially for teosinte, we report and interpret evidence that many sweeps are happening exclusively to one population, which is consistent with the action of location adaptation and consistent with some of our expectations.

More directly, this is another instance of us having clear hypotheses going into the paper and constructing specific analyses to test them. As we explain in the paper, we expected the scale of local adaptation to be very small, such that subspecies growing next to each other have more opportunities to exchange alleles that are locally adapted to their shared environment. The analysis we conducted makes sense in light of this expectation. We considered conducting tests regarding geographic proximity, but there is limited power with the number of populations we have within subspecies, and the meaning of the tests is unclear if all populations of both subspecies are naively included together. This analysis shows that, at least for sweeps and fixations, adaptation is larger than a single location. While it may not be a complete description on its own, the work here does provide information about the scale of adaptation and is useful to our overall claims and objectives of the paper. As mentioned in the paper, the story might be very different if we were to study through a lens of polygenic adaptation. We also now include in the discussion in several places mention of where broader sampling could improve inference.

(4) Convergent adaptationMy biggest concern involves the apparent main conclusion of the paper about the sources of "convergent adaptations". I believe the authors are misapplying the method of Lee and Coop (2017), and have not seriously considered the confounding factors of this method as applied. I am unconvinced by the conclusions that are made from these analyses.The method of Lee and Coop (referred to as rdmc) is intended to be applied to a single locus (or very tightly linked loci) that shows adaptation to the same environmental factor in different populations. From their paper: "Geographically separated populations can convergently adapt to the same selection pressure. Convergent evolution at the level of a gene may arise via three distinct modes." However, in the current paper, we are not considering such a restricted case. Instead, genome-wide scans for sweep regions have been made, without regard to similar selection pressures or to whether events are occurring in the same gene. Instead, the method is applied to large genomic regions not associated with known phenotypes or selective pressures.I think the larger worry here is whether we are truly considering the "same gene" in these analyses. The methods applied here attempt to find shared sweep regions, not shared genes (or mutations). Even then, there are no details that I could find as to what constitutes a shared sweep. The only relevant text (lines 802-803) describes how a single region is called: "We merged outlier regions within 50,000 Kb of one another and treated as a single sweep region." (It probably doesn't mean "50,000 kb", which would be 50 million bases.) However, no information is given about how to identify overlap between populations or sub-species, nor how likely it is that the shared target of selection would be included in anything identified as a shared sweep. Is there a way to gauge whether we are truly identifying the same target of selection in two populations?The question then is, what does rdmc conclude if we are simply looking at a region that happened to be a sweep in two populations, but was not due to shared selection or similar genes? There is little testing of this application here, especially its accuracy. Testing in Lee and Coop (2017) is all carried out assuming the location of the selected site is known, and even then there is quite a lot of difficulty distinguishing among several of the non-neutral models. This was especially true when standing variation was only polymorphic for a short time, as is estimated here for many cases, and would be confused for migration (see Lee and Coop 2017). Furthermore, the model of Lee and Coop (2017) does not seem to consider a completed ancestral sweep that has signals that persist into current populations (see point 3 above). How would rdmc interpret such a scenario?Overall, there simply doesn't seem to be enough testing of this method, nor are many caveats raised in relation to the strange distributions of standing variation times (bimodal) or migration rates (opposite between maize and teosinte). It is not clear what inferences can be made with confidence, and certainly the Discussion (and Abstract) makes conclusions about the spread of beneficial alleles via introgression that seem to outstrip the results.

We have fixed the “50,000 Kb” typo.

There are several important points the reviewer makes here worth considering. First and most importantly, the method of Lee and Coop (2017) actually does include sites as part of the composite likelihood calculation. For computational feasibility, the number of positions we initially considered was 20 (20 different positions along the input sequence were proposed as the site of the shared beneficial mutation). In efforts to further address the reviewer’s concern about adaptive mutations at distinct loci, we have increased the number of proposed selected sites to 200. This fact should greatly diminish the reviewer’s concern that we are picking up independent sweeps that happened at different nucleotide positions in the same region － evidence for a beneficial mutation must be shared by the selected populations at a proposed site. As the revisions show, this has modified the results of our paper in a number of ways, including changing all of the previous neutral regions to shared via standing variation or migration. Despite these changes, our previous conclusions are intact, including the pattern that migration rates are high when maize populations share the sweep. Relatedly, we disagree with the reviewer’s characterization of the migration results. The pattern is quite clear and makes sense － when a maize population is involved in the sweep, migration rate is inferred to be high. Sweeps exclusive to teosinte are rarer and are inferred to have a low migration rate. This relates directly to the idea that humans have moved maize relatively rapidly across the landscape.

We have now included a plot showing how the difference between the maximum composite likelihood (CLE) site compares to the next highest CLE site varies across our inferences (Figure S8), which strongly suggests that patterns are not muddled across multiple loci, but are centered at a focal region where the beneficial allele is inferred to be located. While there are too many to show in the manuscript across all sweeps, here is a nice example of what inference looks like for one of the proposed sweep regions.

**Author response image 1. sa2fig1:** 

Furthermore, the situation the reviewer is describing would be selection acting on independent mutations (mutations at different loci), which would not create an increase in the amount of allele frequency covariance above and beyond what would be expected by drift under the migration and standing variation models.

We also note that we are not alone in applying this approach to shared outlier signals in the absence of known genes; indeed the authors of the DMC method have applied it to regions of shared outlier signal themselves (e.g. https://journals.plos.org/plosgenetics/article?id=10.1371/journal.pgen.1008593).

**Reviewer #2 (Public Review):**
Summary:The authors sampled multiple populations of maize and teosinte across Mexico, aiming to characterise the geographic scale of local adaptation, patterns of selective sweeps, and modes of convergent evolution between populations and subspecies.Strengths & Weaknesses:The population genomic methods are standard and appropriate, including Fst, Tajima's D, α, and selective sweep scans. The whole genome sequencing data seems high quality. However, limitations exist regarding limited sampling, potential high false-positive sweep detection rates, and weak evidence for some conclusions, like the role of migration in teosinte adaptation.Aims & Conclusions:The results are interesting in supporting local adaptation at intermediate geographic scales, widespread convergence between populations, and standing variation/gene flow facilitating adaptation. However, more rigorous assessments of method performance would strengthen confidence. Connecting genetic patterns to phenotypic differences would also help validate associations with local adaptation.Impact & Utility:This work provides some of the first genomic insights into local adaptation and convergence in maize and teosinte. However, the limited sampling and need for better method validation currently temper the utility and impact. Broader sampling and connecting results to phenotypes would make this a more impactful study and valuable resource. The population genomic data itself provides a helpful resource for the community.Additional Context:Previous work has found population structure and phenotypic differences consistent with local adaptation in maize and teosinte. However, genomic insights have been lacking. This paper takes initial steps to characterise genomic patterns but is limited by sampling and validation. Additional work building on this foundation could contribute to understanding local adaptation in these agriculturally vital species.

We appreciate the reviewer’s thoughtful reading of the paper and scrutiny. We hope that the added caveats made in response to reviewer 1 (as well as the previous rounds of peer review) will provide readers with the proper amount of skepticism in the accuracy of some of our initial sweep results, while also demonstrating that many of our conclusions are robust to the concerns raised over the various stages of review.

We agree with the reviewer that better sampling and the incorporation inference about phenotypic data would be excellent additions, but the information is not available for the studied populations, and is outside scope of this paper.

**Recommendations for the authors:**

**Reviewer #1 (Recommendations For The Authors):**
- Sometimes alpha is described as a rate, and sometimes as a proportion. The latter is correct.

We have updated this. Thanks.

- Line 79: are they really "discrete" populations?

The teosinte populations sampled are all clearly separated from each other and are physically discrete. The maize population samples came from individual farmer fields. Traditional maize is grown as open-pollinated (outcrossing) populations, and farmers save seed for subsequent generations. An individual farmer’s field thus behaves as a discrete population for our purposes, impacted of course by gene flow, selection, and other evolutionary processes.

- Lines 418-420: "Large genomes may lead to more soft sweeps, where no single mutation driving adaptive evolution would fix (Mei et al. 2018)." I'm not sure I understand this statement. Why is this a property of genome size?

Mei et al. 2018 lay out the logic, but essentially they present data arguing that the total number of functionally relevant base pairs increases with genome size (less than linearly). If true, genomes with a large number of potentially functional bp are more likely to undergo soft sweeps (see theory by Hermisson and Pennings cited in Mei et al. 2018).

- Lines 500-1: selection does not cause one to underestimate effective population sizes. Selection directly affects Ne. I'm not sure what biases the sentences on lines 502-508 are trying to explain.

We have simplified this section. Not accounting for linked selection (especially positive selection) results in a biased inference of demographic history. See Marsh and Johri (2024) for another example. https://doi.org/10.1093/molbev/msae118

- Line 511-3: does Uricchio et al. (2019) show any difference in the estimate of alpha from Messer and Petrov (2013) when taking background selection into account?

What we initially wrote was incorrect. The aMK method of Messer and Petrov (2013) accounts for weakly deleterious polymorphisms, but it does not account for positively selected ones. We have updated this text and suggested our method may underestimate alpha if positively selected segregating alleles are common (near line 539).

- Lines 598-599: "which would limit the rate of new and beneficial mutations." I don't understand this - shouldn't a bottleneck only affect standing variation? Why would a bottleneck affect new mutations?

This is simply to say that during the low Ne period of a bottleneck, fewer total mutations (and therefore beneficial mutations) will be generated since there are fewer individuals for mutations to occur in. We have changed “rate” to amount to clarify we do not mean the mutation rate itself.

**Reviewer #2 (Recommendations For The Authors):**
Experiments/Analyses:(1) Consider simulating polygenic adaptation in addition to hard and soft sweeps to see if this improves the power to detect adaptive signatures shared between populations. This could involve simulating the coordinated change in allele frequencies across many loci to match a specified shift in trait value due to selection. The ability to detect shared polygenic adaptation between population replicates could be assessed using methods tailored to polygenic signals, such as the Polygenic Selection Score approach. Comparing the power to detect shared polygenic adaptation versus shared hard and soft sweeps would provide further insight into what adaptive modes current methods can uncover. If the power to detect shared polygenic adaptation is very low, the extent of shared adaptation between populations may be even more common than currently inferred. Adding simulations of polygenic adaptation would strengthen the study.

While this would be a worthwhile undertaking in general, it would be a considerable amount of work outside of the scope and aims of this paper.

(2) Explore using machine learning approaches like S/HIC to improve power over summary statistic methods potentially.

We in fact put considerable effort into applying diplo S/HIC before switching to raisd for this project. While predictions on simulations had good power to detect sweeps, we found that applying to our actual data had a dubious number of windows classified as sweeps (e.g. >90% of the genome), which we believed to be false positives. We speculated that this may have to do with sensitivity to demographic or other types of misspecification in the simulations, such as our choice of window sizes compared to local recombination rates. It would likely be fruitful to our further efforts into using machine learning methods for maize and teosinte, but a deeper exploration of the right hyper parameters and simulation choices is likely needed to apply them effectively.

(3) Increase geographic sampling density, if possible, especially near population pairs showing high differentiation, to better understand the scale of local adaptation.

We agree this would be valuable research. Hopefully this work inspires further efforts into the question of the spatial and temporal scales of local adaptation with more ambitious spatial sampling designed at the onset

Writing/Presentation:(1) Provide more intuition about the biological interpretation of the migration rates inferred under the migration model of convergence. What do the rates imply about the amount or timing of gene flow?

We have expanded the discussion sections (starting near line 653) to elaborate on the migration results and connect the rdmc and f4 tests more explicitly. The timing of gene flow is more challenging to address directly with the approaches we used, but we agree it would be interesting to explore more in future papers.

(2a) Expand the discussion of power limitations and the need for simulation tests. Consider adding ROC curves for sweep detection on simulated data. The relatively low proportion of shared selective sweeps between population replicates highlights limitations in the power to detect sweeps, especially incomplete or soft sweeps. I think it would be a good idea to expand the discussion of the power tradeoffs shown in the simulation analyses. In particular, the ROC curves in Figure S4 clearly show how power declines for weaker selection coefficients across the different sweep types. I suggest making these ROC curves part of the main figures to feature the issue of power limitations more prominently.(2b) The discussion would benefit from commenting on how power changes across the sweep simulation scenarios. Adding a summary figure to visualise the effects of sweep type, selection strength, and frequency on detectability could further clarify the power constraints. Stating the proportion of sweeps likely missed strengthens the argument that sharing adaptive alleles is likely even more common than inferred. Discussing power will also motivate the need for developing methods with improved abilities to uncover incomplete and soft sweeps.

While these are useful suggestions (2a and 2b), the aim of this paper at its core is empirical, and was not intended to give an exhaustive analysis of the power to detect sweeps. We report what parts of the analysis may be impacted by low power and what aspects of our inferences have higher uncertainty due to power. We agree that there is more work to be done to improve methods to detect selection given our findings (see below concerning our efforts to use machine learning as well). While we do not highlight this in the paper, we also note that ours is one of extremely few empirical studies that actually perform power analyses on real data (as opposed to simulations). We think this extra transparency by itself is of substantial utility to the community in demonstrating that the results from simulation studies performed in publications describing a method do not necessarily translate well to empirical data.

(3) Improve clarity in describing f4 test results. Consider visualising results on a map to show spatial patterns.

We have expanded the discussion concerning f4 tests (see several comments to reviewer 1). We are not clear on how to effectively visualize f4 spatially, but hope the updates have made the results more clear.

Minor:- Increase the font size of figure axis labels for improved readability.

We have looked over and figures and increased font sizes where possible.

- Add units to selection coefficient axis labels in Figure 5.

Selection coefficients are derived in Lee and Coop (2017) from classical population genetics theory. They do not have units, but denote the relative fitness advantage of the heterozygous genotype carrying the beneficial mutation of interest.

- Fix the typo 'cophenetic' in Figure S3 caption.

Fixed. Thank you.